# Dynamic metastable long-living droplets formed by sticker-spacer proteins

**Srivastav Ranganathan, Eugene I Shakhnovich***

Department of Chemistry and Chemical Biology, Harvard University, Cambridge, United States

**Abstract** Multivalent biopolymers phase separate into membrane-less organelles (MLOs) which exhibit liquid-like behavior. Here, we explore formation of prototypical MOs from multivalent proteins on various time and length scales and show that the kinetically arrested metastable multi-droplet state is a dynamic outcome of the interplay between two competing processes: a diffusion-limited encounter between proteins, and the exhaustion of available valencies within smaller clusters. Clusters with satisfied valencies cannot coalesce readily, resulting in metastable, long-living droplets. In the regime of dense clusters akin to phase-separation, we observe co-existing assemblies, in contrast to the single, large equilibrium-like cluster. A system-spanning network encompassing all multivalent proteins was only observed at high concentrations and large interaction valencies. In the regime favoring large clusters, we observe a slow-down in the dynamics of the condensed phase, potentially resulting in loss of function. Therefore, metastability could be a hallmark of dynamic functional droplets formed by sticker-spacer proteins.

## Introduction

Biomolecular phase transitions are widespread in living systems. Transitions that result in irreversible, solid-like assemblies such as amyloid fibrils are a hallmark of disease while those like cytoskeletal filaments play a functional role (*Banani et al., 2017*). The third self-assembled state, in addition to the soluble and the solid-like state, is the loosely associated droplet phase held together by several weak, transient interactions (*Guo and Shorter, 2015*). The transient nature of these interactions makes these self-assemblies reversible and thereby a potential strategy for a temporally regulated sub-cellular organization. Several examples of spatiotemporally regulated, droplet-like objects within the cell have been discovered in the past few decades, composed of different types of proteins often co-localized with nucleic acids (*Hyman et al., 2014*). The importance of these membrane-less compartments is potentially two-fold: (i) localizing biochemical reactions within the cell, and (ii) sequestering biomolecules to regulate their activity (*Hyman et al., 2014*). Examples of cytoplasmic membrane-less organelles include P bodies, germ granules and stress granules (SGs) (*Mitrea and Kriwacki, 2016*). However, aberrant granule dynamics and a transition from a liquid-like to a more solid-like state are often hallmarks of degenerative diseases. (*Kim et al., 2013*; *Ramaswami et al., 2013*) Solid-like RNP aggregates have been reported as cytoplasmic inclusions (*Patel et al., 2015*) and as nuclear RNP aggregates (*Ramaswami et al., 2013*) in degenerative diseases. Investigating the physical principles governing the formation of these high-density phases is vital to understand the subcellular organization and the conditions leading to disease.

 Several experimental studies have highlighted the 'multivalent' nature of the constituent proteins in membrane-less organelles (*Banani et al., 2017*; *Li et al., 2012*; *Xing et al., 2018*). In other words, these proteins carry multiple associative or 'adhesive' domains (*Patel et al., 2015*; *Li et al., 2012*). Multivalency could be achieved by several different architectures (*Banani et al., 2017*), the simplest being a linear sequence of folded domains that are connected by linker regions. Li et al., employed a linear multivalent 2-component model system (SH3 and PRM domains threaded together by

***For correspondence:**
shakhnovich@chemistry.harvard.edu

**Competing interests:** The authors declare that no competing interests exist.

flexible regions) to show that liquid-like droplet formation can result from just two multivalent interacting components (repeats of the same domain) (*Li et al., 2012*). Another intriguing feature of condensate proteins is the presence of intrinsically disordered regions or low complexity sequences that link folded domains together (*Harmon et al., 2017*). Therefore, a combination of these motifs and different architectures could form a basis for different types of phase-separated structures within the cell. In this paper, we address a specific question – how does this multi-domain sticker-spacer architecture shape the dynamics of droplet formation and further growth? Previous computational studies, notably the coarse-grained simulations by Harmon et al., address the nature of the equilibrium gel-like state formed by multivalent polymers with a sticker-spacer architecture (*Harmon et al., 2017*; *Choi et al., 2019*). The equilibrium phase diagrams for the 2-component SH3-PRM system and the effect of regulator molecules have also been studied using patchy particle models by Zhou and co-workers (*Ghosh et al., 2019*; *Nguemaha and Zhou, 2018*). Theoretical studies have also employed the Flory theory of phase separation in polymer solutions to describe the thermodynamics of membraneless organelle (*Brangwynne et al., 2015*) formation. The Flory theory, which applies to solutions of homopolymers predicts two phases – fully mixed and a single, large phase-separated droplet. Equilibrium theories robustly establish the underlying two-state equilibrium landscape that drives biopolymers to localize into 'polymer-rich' phases within the cell (*Harmon et al., 2017*; *Choi et al., 2019*; *Brangwynne et al., 2015*; *Flory, 1942*). Indeed, the growth of liquid droplets via the mechanisms of Ostwald ripening, coalescence or by consumption of free monomers from the surrounding medium (at concentrations greater than the saturation threshold) has been observed in vitro (*Guillén-Boixet et al., 2020*; *Zhang et al., 2015*; *Martin et al., 2020*). However, in a deviation from the equilibrium picture, membraneless organelles such as the nucleoli and P bodies are also known to exist in the form of multiple co-existing droplets (with equilibrium-like droplet environment) in vivo at biologically relevant time-scales (*Brangwynne et al., 2011*; *Kilchert et al., 2010*; *Berciano et al., 2007*). (*Dine et al., 2018*) report the persistence of multiple optogenically controlled droplets over long periods of time in vitro, a finding that is attributed to a progressive slow-down over time of droplet growth via Ostwald ripening. Active ATP-dependent processes have been attributed to the coexistence of multiple droplet phases, and the regulation of their size distributions in the cell (*Wurtz and Lee, 2018a*; *Weber et al., 2019*; *Zwicker et al., 2017*). Other potential factors hindering droplet fusion could be structural rigidity of the droplet scaffold (*Boeynaems et al., 2019*) and the aging of droplets due to internal re-organization (*Lin et al., 2015*). However, the potential existence of a more general, ATP-independent mechanism resulting in the prevalence of a stable multi-droplet system over biologically relevant time-scales (*Wegmann et al., 2018*; *Li et al., 2012*) has not been explored yet.

Despite significant experimental and computational efforts, a key question remains unanswered – what are the physical mechanisms that result in a long-living, metastable, multi-droplet system (*Wegmann et al., 2018*; *Dine et al., 2018*) in the absence of active processes? How does the sticker-spacer architecture of self-assembling biopolymers, with finite valencies, contribute to the slow-down and arrested coalescence of the early droplets into a single macro-phase? Since the equilibrium states in such systems are either mixed or a single, large droplet state, the kinetic factors must play a crucial role in the prevalence of the metastable multi-droplet state (*Li et al., 2012*; *Dine et al., 2018*). Also, if the system of multiple droplets is metastable, what are the physical factors that govern the kinetics of cluster growth? Measurable quantities such as droplet sizes also assume importance in the context of biological function, where the size of the membraneless organelles has been linked to function (*Brangwynne, 2013*). It is, therefore, not just the tendency to phase separate but also the size of these assemblies that must be regulated for their proper functioning (*Goehring and Hyman, 2012*). In this context, understanding the potential molecular mechanisms tuning the dynamics of cluster growth at biologically relevant time-scales becomes extremely relevant.

In this work, we address these questions using multi-scale coarse-grained models. In the first part of the paper, we probe the physical determinants of the formation of long-living metastable micro-droplets using a coarse-grained model of multivalent polymers composed of a linear chain of adhesive domains separated by semi-flexible linkers. The results of our Langevin dynamics (LD) simulations for this model shed light on the early stages of droplet growth and the mechanism of arrested phase-separation. Next, using the LD simulations as the basis to identify vital time-scales for condensate growth, we explore the phenomenon at biologically relevant time-scales using a

phenomenological kinetic model. Broadly, we explore the kinetically arrested liquid-liquid phase separation (LLPS) on multiple scales – from mesoscale to macroscopic. Overall, these results provide a detailed mechanistic understanding of the factors that determine the prevalence of the metastable multi-droplet state for spacer-sticker heteropolymers in the absence of active, ATP-dependent processes. While the thermodynamic landscape drives phase-separation, the crucial role played by dynamic processes could result in measurable quantities deviating from the equilibrium predictions. The current study sheds light on the importance of dynamics of cluster growth, complementing the existing equilibrium understanding of the phenomenon.

## Model

Despite the complexity of the intracellular space, experiments suggest that in vitro liquid-liquid phase separation (LLPS) can be achieved even using simple two-component systems (*Li et al., 2012*). The tractability of simpler models makes them powerful tools to investigate the role of physical factors in modulating droplet formation and growth. Here, we perform LD simulations (see Materials and methods for detail) to understand the process of self-association between two types of polymer chains composed of specific interaction sites (red and yellow beads in *Figure 1*). These adhesive sites are linked together by non-specifically interacting linkers (blue beads in *Figure 1*). The red and yellow beads on these chains mimic complementary domains on different chains that can participate in a maximum of one specific interaction (between yellow and red beads).

For our simulations, we consider 400 such semi-flexible polymer chains (200 of each type) in a cubic box with periodic boundary conditions). Each chain in the simulation box is composed of 5 specific interaction sites that are linked together by non-specific linker regions that are 35 beads long. (blue beads in *Figure 1*). This linker length was based on previous theoretical studies of phase-separating proteins (*Harmon et al., 2017*). We employ conventional Langevin dynamics simulations to study the self-assembly of the model biopolymers, wherein the size of the specific interaction sites

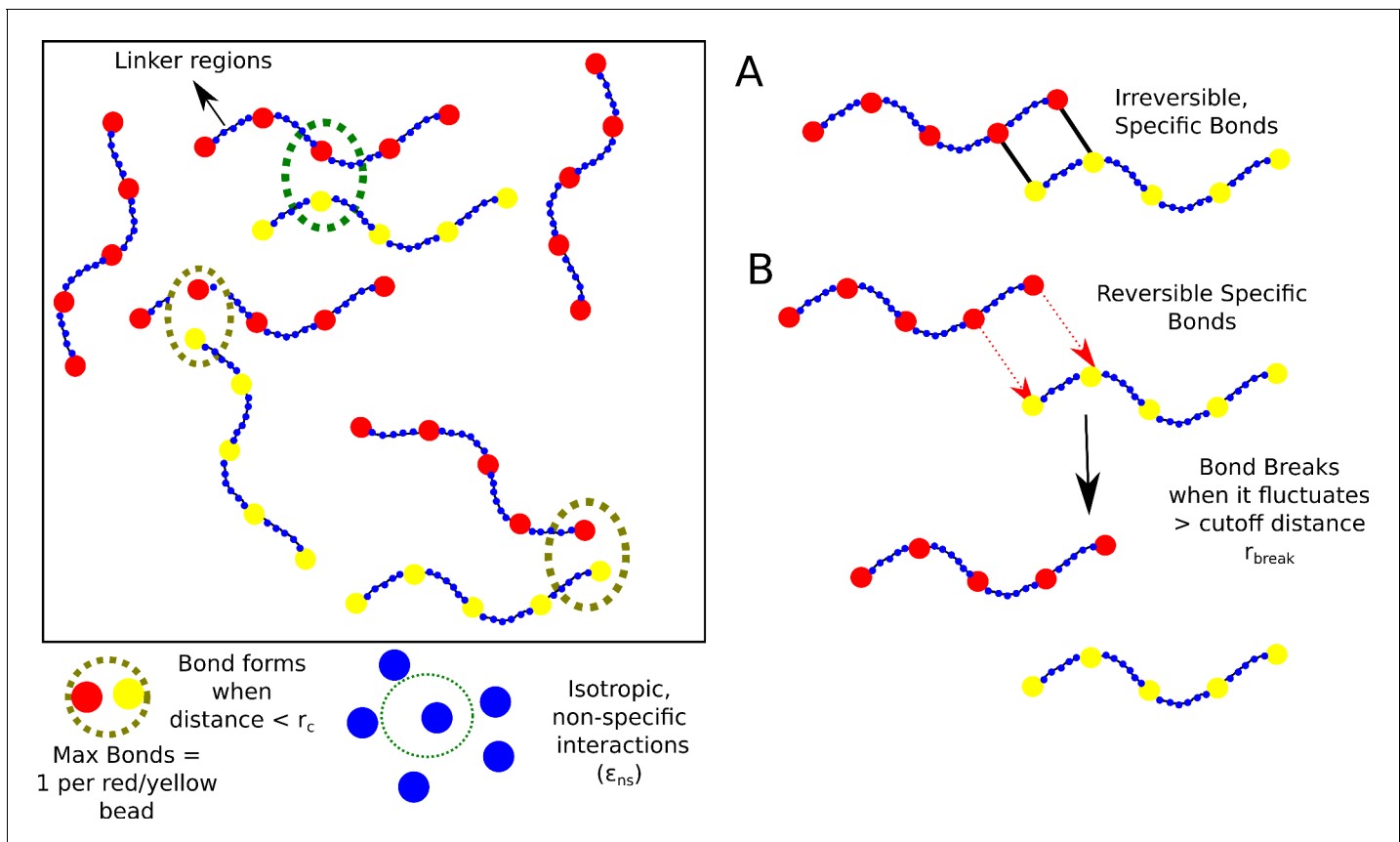

**Figure 1.** Model. Schematic of the polymer model for studying phase-separation by multivalent biopolymers.

(diameter of 20 Å) is roughly four times that of the linker beads which represent individual amino acid residues (diameter of 4.2 Å). This difference in sizes is to mimic a folded adhesive domain – SH3 domain (diameter of 20 Å, PDB ID:1SHG,) that is often involved in liquid-liquid phase separation (*Musacchio et al., 1992*). The folded adhesive domains were modelled at a lower resolution (one bead per domain of 60 amino acids) than linkers which are more dynamic and hence modelled with a one bead per amino acid resolution.

## Results

### Terminology and order parameters used in this study

Self-assembly of multivalent polymers with inter-chain interactions between complementary domains is a simple model system for studying phase-separation by intracellular polymers. In the current study, we employ this model to discover microscopic factors driving phase separation. In the first part of this study, we present results of self-assembly driven by irreversible (highly stable) functional interactions and identify two key time-scales that define cluster growth and their size distributions using Langevin dynamics (LD) simulations. In the latter part of this paper, we build a coarse-grained kinetic model demonstrating the tunability of this phenomenon using kinetic Monte Carlo (kMC) simulations. In the following subsections, we use the term specific interactions for the finite valency interactions between functional domains and non-specific interactions to refer to the inter-linker isotropic interactions. It must be noted that the emphasis of our simulations is to shed light on the factors that suppress the coalescence of individual clusters into a single, large equilibrium-like cluster. Also, the focus of the current study is to understand the process of cluster growth and the physical mechanisms modulating measurable quantities such as droplet size distributions at the early (LD) and biologically relevant time-scales (kMC). In *Table 1*, we summarize the key simulation variables and order parameters used in this study.

When $L_{clus}$—> $N_{tot}$, we refer to the state as a system-spanning 'macro-phase'. A system of multiple, co-existing clusters with with $S_{clus}$<< $N_{tot}$, is referred to as a metastable 'micro-phase'. It is important to note that the use of the term 'micro-phase' in this paper is in reference to metastable, and not equilibrium phases. In order to establish the condensed nature of micro-phase clusters as compared to the system-spanning macro-phase, we characterize the intra-cluster densities normalized by the density of a system ($\phi_{clus}$ /$\phi$) of randomly placed polymer chains in the simulation box. We use an analogous term for density measurements in our kinetic Monte Carlo simulations, with the volume term being replaced by the area of the cluster (or lattice) in 2D. For details of the potential functions and the implementation of these simulations, please refer to the Materials and methods section.

### Metastable 'micro-phases' – a likely outcome at low to intermediate concentrations

Multivalent polymers with adhesive domains separated by flexible linkers can potentially self-assemble through two types of interactions, a) the finite number of adhesive contacts between the functional domains (yellow and red beads in *Figure 1*), and b)non-specific, isotropic interactions between the linker regions (blue beads in *Figure 1*). We first performed control simulations with specific interactions turned off, where we varied free monomer concentration $C_{mono}$, from 10 to 200 μM for a weak non-specific interaction strength of $\epsilon_{ns}$ = 0.1 kcal/mol. In these control simulations, we observe no phase-separation in the whole range of $C_{mono}$(*Figure 2—figure supplement 1A*, purple curve) . Therefore, in this regime of $\epsilon_{ns}$ and $C_{mono}$, at the simulation time-scale of 16 μs, the polymer assemblies do not reach large sizes. Not surprisingly, this result suggests that the assembly driven by non-specific interactions (linker-driven) alone is achieved only at stronger non-specific interactions and/or high free monomer concentrations. However, intracellular phase-separation often results in the enrichment of biomolecules within the self-assembled phase, at relatively low protein concentrations (*Xing et al., 2018*). Under such conditions, specific interactions which are fewer in number become critical determinants of phase separation. In our LD simulations, we employ polymer chains comprised of 5 univalent adhesive domains and four linker regions totalling 145 beads, bringing a total valency of 5 for each polymer. For the sake of simplicity, we begin with a situation where these bonds, once formed, do not break for the rest of the simulation. Although an idealized construct

**Table 1.** Important simulation variables and order parameters.

| Notation/ Terminology | Physical Interpretation | Definition |
|---|---|---|
| $N_{tot}$ | Total number of polymer chains in the system. $N_{tot}$ = 400 in our simulations. | – |
| $L_{clus}$ | Size of the single largest cluster | represented as fraction of $N_{tot}$ |
| $S_{clus}$ | Size of the cluster | represented as fraction of $N_{tot}$ |
| $C_{mono}$ | Concentration of polymer chains in the simulation box | In units of μM |
| φ | Bulk density of proteins (in their monomeric state) when the individual chains are randomly placed in the simulation box at the start of the simulation. | $\phi = \frac{N_{tot}}{(4/3)\pi R_G^3} . R_g'$ is the the radius of gyration of proteins when they are randomly positioned in simulation box at the start of the simulation. |
| $φ_{clus}$ | Intra-cluster density of polymer chains. | $\phi_{clus} = \frac{S_{clus}}{(4/3)\pi R_g^{3^{clus}}} . R_g^{clus}$ is the radius of gyration of the system of proteins within the cluster. |
| $φ_{clus}$ /φ | Normalized intracluster density describing the degree of enrichment of polymer chains within the cluster. | For system-spanning networks, $φ_{clus}$ /φ→1. For dense clusters, $φ_{clus}$ /φ >> 1. |
| $ε_{ns}$ | Interaction strength for isotropic, non-specific interactions between linker regions. $ε_{ns}$ in LD simulations is a pairwise interaction strength between individual beads In kMC simulations, $ε_{ns}$ is the net non-specific interaction strength between two lattice particles. | – |
| $ε_{sp}$ | Strength of attractive interaction between functional domains (specific interactions). | – |
| $φ_{lattice}$ | Bulk density of monomers in the 2D-lattice in kMC simulations. This quantity is analogous to the concentration of monomers on the lattice. | $φ_{lattice}$ = $N_{tot}/L^2$, where L is the size of the 2D square-lattice. |
| λ | Valency of the polymer chain, that is the number of adhesive functional domains per interacting polymer chain. | – |
| K | Bending stiffness of the linker regions in LD simulations. A higher value of K is used to model stiffer linkers that prefer a more open configuration. | K = 2 kcal/mol, in LD simulations with flexible linkers. |
| η | The viscosity of the medium in LD simulations | η = $10^{-3}$Pa.s in LD simulations, unless mentioned otherwise. |
| $k_{diff}$ | Diffusion rate of free monomers in the 2D-lattice kMC simulations. | – |
| $k_{bond}$ | Rate of formation of specific interactions between neighboring particles in 2D-lattice kMC simulations. | – |

with respect to biological polymers, these simulations can build intuition about cluster size distributions and arrest of cluster growth when the spontaneously assembled clusters are unable to undergo further re-organization.

In *Figure 2A* we show the single largest cluster size, for varying free monomer concentrations ($C_{mono}$). The polymer-chains, for the data plotted in *Figure 2*, have flexible linkers ($κ$=2 kcal/mol, see *Table 1*. and Materials and methods for definition of bending rigidity) with weak inter-linker interactions ($\epsilon_{ns}$=0.1 kcal/mol, see *Table 1*. And Materials and methods section for definition). As evident from *Figure 2*, the polymer chains can assemble into larger clusters in the presence of functional interactions (black curve, *Figure 2A*), as opposed to the control simulations where inter-linker interactions alone drive self-assembly (*Figure 2—figure supplement 1A*, purple curve). Simulations with irreversible functional interactions suggest that, for low and intermediate concentrations ($C_{mono}$<70 μM), $S_{clus}$<< $N_{tot}$. Also, at these concentrations, we observe a distribution of cluster sizes (*Figure 2C*, blue bars) indicating a multi-cluster 'microphase' state. We also characterized the density of the clusters using a second order parameter – $\phi_{clus}$ /$\phi$. This quantity is analogous to the measure of phase separation employed in a previous study by *Harmon et al., 2017*. The density transition as a function of concentration shows a non-monotonic behavior. For large $C_{mono}$, where the mean largest cluster sizes approach the size of the system, we do not observe a polymer-dense phase. In this regime (pink region in *Figure 2A*, where $\phi_{clus}/\phi \to 1$, the self-assembled state is a percolated, system spanning network. In a narrow intermediate regime of concentration (cyan region in *Figure 2A*), we

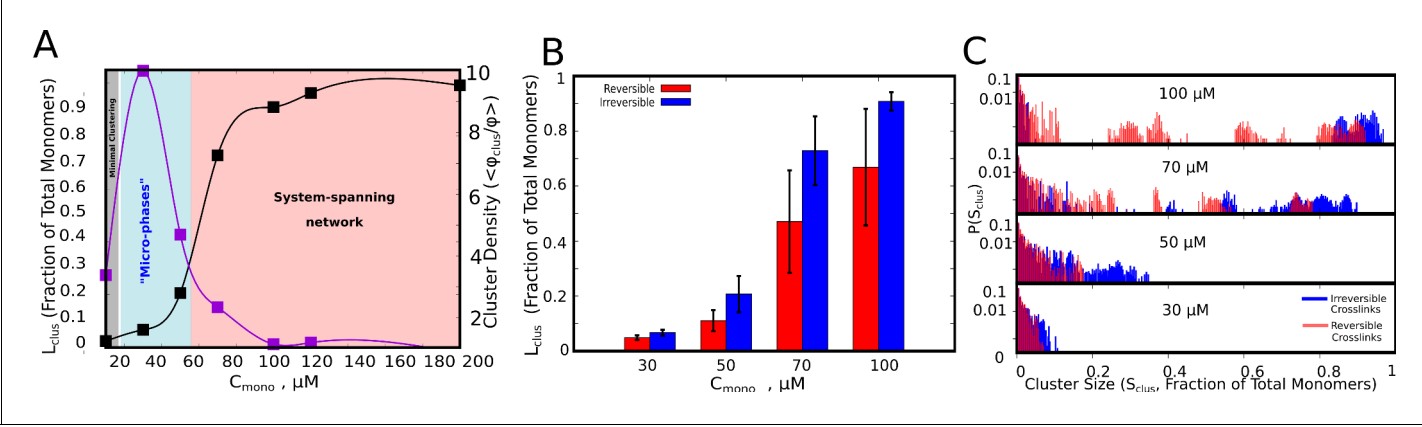

**Figure 2.** Cluster Sizes in Langevin Dynamics simulations. (**A**) Black Curve – The single largest cluster as a function of free monomer concentration (in μM) for irreversible functional interactions. The largest cluster size is shown as a fraction of the total number of monomers in the simulation box. Purple Curve – The mean density of the cluster, $\phi_{clus} = \frac{S_{clus}}{(4/3)\pi R_g^{3,clus}}$, normalized by the bulk density of monomers in the simulation, $\phi = \frac{S_{clus}}{(4/3)\pi R_g^{3,r}}$. $S_{clus}$ and $N_{tot}$ refer to the size of the cluster and the total number of monomers in the simulation box, respectively. $R_g^{clus}$ and $R_g^r$ refer to the radius of gyration of the cluster, and the radius of gyration of proteins in their randomly located initial configuration at the start of the simulation, respectively. The quantity $\phi_{clus}/\phi$ shows the degree of enrichment of polymer chains within the cluster upon self-assembly. The smooth curves are plotted as a guide to the eye, using the cspline curve fitting. (**B**) Comparison of the mean sizes of the single largest cluster for reversible and irreversible specific interactions, for varying free monomer concentrations. (**C**) Cluster size distributions for varying free monomer concentrations for reversible and irreversible specific interactions. Darker shades of red in the distributions indicate verlapping regions of the distribution while the blue and light red shades indicate regions with no overlap in the presence and absence of breakable interactions. The linker stiffness for the self-assembling polymer chains in this plot is 2 kcal/mol while the strength of inter-linker interaction is 0.1 kcal/mol (per pair of interacting beads). The mean and distributions of the largest cluster sizes were computed using 500 different configurations from five independent simulation runs of 16 μs.

The online version of this article includes the following source data and figure supplement(s) for figure 2:

**Source data 1.** Compressed zip file containing the source data for cluster sizes and cluster size distributions (along with raw unprocessed data) plotted in *Figure 2*.

**Figure supplement 1.** Cluster Sizes in Langevin Dynamics simulations.

observe microphases with high polymer density within the cluster. The density transitions are qualitatively consistent with predictions from equilibrium theories for sticker-spacer proteins studied by *Harmon et al., 2017*. Although the density transitions are in tune with the equilibrium landscape, the key prediction of our simulations is the microphasic nature of this high-density regime with coexistence of multiple clusters (*Figure 2A and C*). The intra-cluster density, however, would also depend on the characteristics of the linker, a phenomenon we demonstrate in subsequent sections.

To further test whether this early time-scale behavior changes in the presence of reversible interactions, we introduced breakable specific bonds in our model (see Materials and methods section for details of the implementation). As with the irreversible interaction simulations, even in the presence of breakable interactions, $L_{clus} << N_{tot}$ (*Figure 2B*) except for large $C_{mono}$ when we observe a system-spanning network. Critically, we find coexistence of intermediate cluster sizes with small and large clusters suggesting an increased diversity in cluster sizes at the early stages of droplet assembly upon introduction of breakable interactions (*Figure 2C*).

Overall, these results suggest that in the concentration regime of dense clusters (*Figure 2A*, blue shaded region), we observe a distribution of cluster sizes (*Figure 2C*) as opposed to a single, large cluster. Given that the cellular concentration of phase-separating proteins is often in the nanomolar to the low micromolar range (*Xing et al., 2018*), the multi-droplet state could persist in vitro and in vivo even in the absence of active processes. In the subsequent sections, we explore the mechanisms that could prevent or significantly delay the coalescence of multiple small clusters into a single large equilibrium-like cluster, eventually resulting in a distribution of droplet sizes.

## Exhaustion of free valencies results in kinetically arrested droplets

Our LD simulations so far reveal an interesting trend – while the intracluster density transition shows a non-monotonic dependence on concentration that is qualitatively consistent with equilibrium theoretical predictions (*Harmon et al., 2017*), we do not observe a single, large equilibrium-like cluster at low concentrations. Crucially, in the regime where the intra-cluster densities are at their highest, we observe a distribution of cluster sizes (*Figure 2A and C*), with $S_{clus} << N_{tot}$, suggesting the prevalence of arrested micro-phases. Therefore, identifying the time-scales which are vital for cluster growth could reveal the cause of arrested droplets growth. In *Figure 3A and B*, we show the time

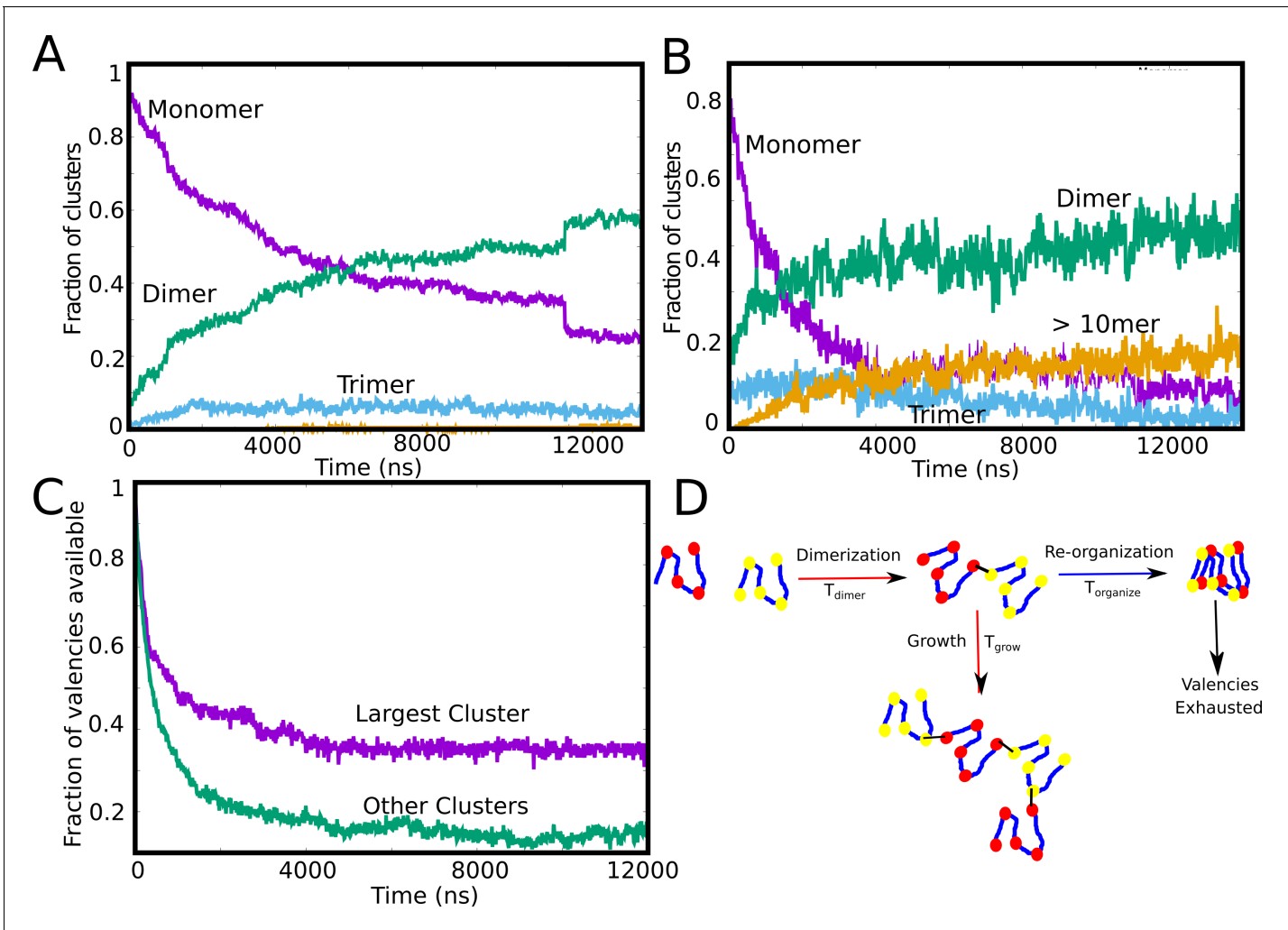

**Figure 3.** Tracking cluster formation at early timescales. **A** and **B** show the temporal evolution of specific contacts for a free monomer concentration of 10, and 50 μM, respectively. For a low concentration of 10 μM, there is an initial decrease in the monomer population (purple curve) which is concomitant with an increase in the dimer population. A negligible fraction of the clusters is in the form of large-mers (size > = 10, orange curve) at these low concentration since the available valencies for growth are consumed by the smaller aggregated species. An increase in concentration from 10μM to 50μM results in an increase in the large-mer (orange curve, 50 μM) population as the monomer fraction decreases during the simulation. A higher free monomer concentration allows the larger clusters to grow due to consumption of free monomers (with unsatisfied valencies) before they get converted into smaller clusters (dimers, trimers) with satisfied valencies. (C) Time evolution of available valencies within the single largest cluster and outside the single largest cluster, for a $C_{mono}$ of 50 μM, $\epsilon_{ns}$ of 0.1 kcal/mol and linker bending rigidity of 2 kcal/mol. (D) A schematic figure showing the possible mechanisms of cluster growth and arrest and the competing timescales that could punctuate the process.

The online version of this article includes the following source data and figure supplement(s) for figure 3:

**Source data 1.** Compressed zip file containing the source data for temporal evolution of different species (dimer,trimer,monomer etc) for different concentrations plotted in *Figure 3*.

**Figure supplement 1.** The size of the largest cluster, for different values of bond formation probability, $P_{form}$.

evolution of the individual aggregate species at different $C_{mono}$ (10μM and 50 μM). As seen from *Figure 3A and B*, for irreversible functional interactions, the monomer fraction continues to monotonically decrease during the simulations with the fraction of other competing species increasing concomitantly. However, at low concentrations (*Figure 3A*, 10 μM), the monomer fraction curve (*Figure 3A*, purple curve) shows a cross-over with the dimer curve (*Figure 3A*, green curve) while higher-order clusters do not appear at simulation time-scales. This result suggests that the spontaneous formation of large assemblies held together by functional interactions is contingent upon two time-scales. The first is the diffusion-limited time-scale that governs initial dimerization and the subsequent growth of these smaller clusters. The second, competing time-scale is the one where all functional valencies get exhausted within the smaller initial clusters. At an intermediate concentration of 50μM, (*Figure 3B*) we observe that the fraction of large aggregates (>10 mer) increases during the early part of the simulations before free monomers are entirely consumed within dimers. In other words, the unsatisfied valencies within the monomers, dimers and trimers get utilized to form larger-sized clusters before the specific valencies get exhausted within the smaller clusters making them no longer available for further self-assembly. To further verify this observation, we track the temporal evolution of the fraction of available valencies within and outside the single largest cluster (*Figure 3C*). We observe that while there are unutilized valencies within the single largest cluster (*Figure 3C*, purple curve), the cluster does not grow further due to almost complete exhaustion of valencies within the polymer chains outside the single largest cluster (*Figure 3C*, green curve). A slower rate of exhaustion of free-valencies within micro-clusters by varying the specific bond formation probability, $P_{form}$ , results in larger cluster sizes for smaller free monomer concentrations (*Figure 3—figure supplement 1*), suggesting that the dynamicity of bond formation can alter the cluster size distributions significantly. These results suggest that the ability to form a single, large, macro-cluster is limited by the exhaustion of free valencies within smaller sized clusters, thereby arresting their growth (*Figure 3D*).

## Identifying the vital timescales determining cluster growth

For stable functional interactions, the process of phase separation gets arrested due to kinetically trapped clusters which do not participate in further cluster growth due to lack of available valencies (*Figure 3D*). As discussed above, two critical time-scales would then dictate the growth of clusters: i) the time-scale for two chains to meet and form the first functional interaction, and ii) the time it takes for the polymer chains within an assembly to exhaust all valencies within the cluster before new chains join in. We explore the factors that these two timescales depend on, using the primary unit of any self-assembly process – the dimer. Using LD simulations, we first compute the mean first passage times for two polymer chains to form their first functional interaction. Given the diffusion-limited nature, this time-scale varies inversely with the concentration of free monomers (represented in the form of density) in the system (*Figure 4A*).

This diffusion-limited time-scale dictates the encounter probability of the two chains. Once the first bond is formed, resulting in an 'active dimer' (one that still has unsatisfied valencies), the cluster can only grow to larger sizes as long as the dimer remains active. Therefore, the time taken by the dimer to exhaust all its valencies becomes a vital second time-scale. In *Figure 4B*, we plot the mean first passage times for a dimer to exhaust all its valencies once the first bond is formed. For low linker stiffness ($κ$<2 kcal/mol in *Figure 4B*), the linkers behave like flexible polymers. In this limit ($κ$<2 kcal/mol), the mean time for the valency to exhaust within the dimer (referred to as the re-organization time, $T_{exhaust-valency}$ ) is independent of linker stiffness. For ($κ$>2 kcal/mol), the linkers behave like semi-flexible polymers and the re-organization time shows a dependence on the stiffness of the linker, with an increase in $T_{exhaust-valency}$ with $κ$. As the linker region becomes more rigid, this time-scale becomes slower, resulting in the dimer remaining 'active' for much longer. A slower re-organization time and a faster diffusion encounter time favors the cluster growing into larger sizes. It must however be noted that an increase in $T_{exhaust-valency}$ via increased linker rigidity would not only influence the cluster sizes but also the intracluster density. For stiffer linkers (and also more open linker configurations), the self-assembled state would approach a system-spanning network as shown in previous studies with linkers with greater effective solvation volumes (*Harmon et al., 2017*). Conversely, a faster re-organization time that is of the order of the encounter time-scales results in a

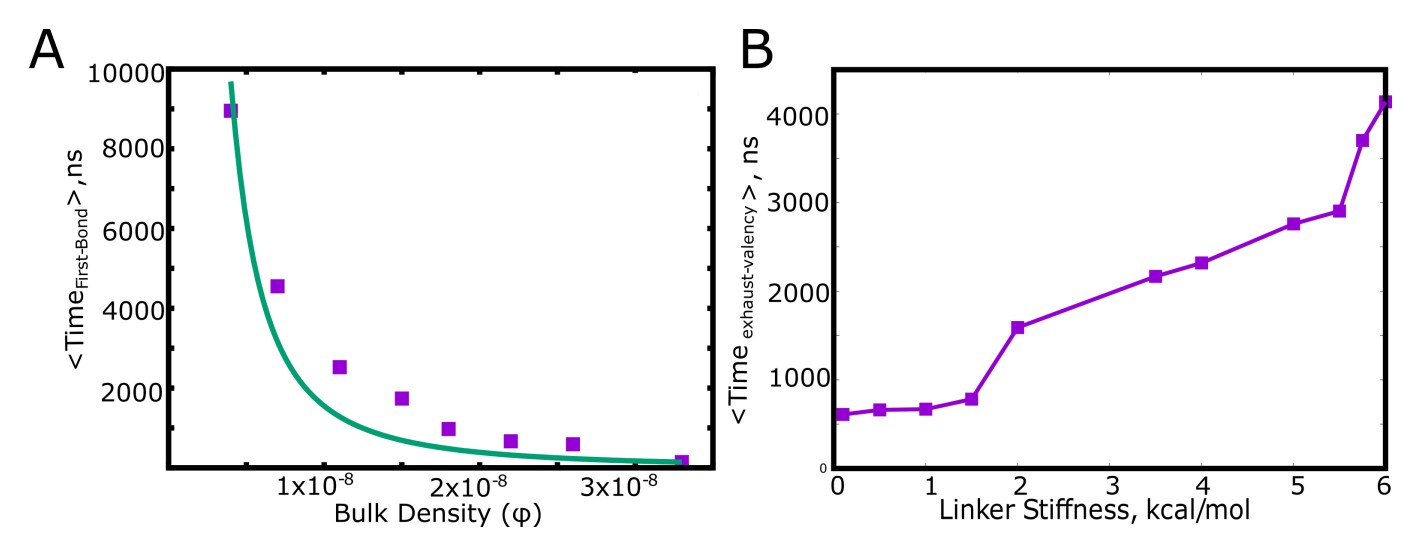

**Figure 4.** Factors influencing the key timescales. (A) The mean first passage time for the first specific interaction between a pair of polymer chains as a function of the bulk density, $\phi$, (see **Table 1** for definition). (B) The mean first passage time for a pair of polymer chains to exhaust all the available valencies within the dimer.

The online version of this article includes the following source data for figure 4:

**Source data 1.** Compressed zip file containing the source data for **Figure 4**.

higher likelihood of the clusters getting locked at smaller sizes. For stable functional interactions, these two time-scales would dictate the size of the largest cluster.

## Linker regions as modulators of self-assembly propensity

The spatial separation of specifically interacting domains (with finite valencies) and the non-specific linker regions is an architecture that is highly amenable to being tuned for phase-separation propensities. In this context, we further extend the findings from our LD simulations to probe how the microscopic properties of the linker region could modulate the extent of phase-separation. In *Figure 5*, we demonstrate how the linker properties could be useful modulators of cluster sizes, without any alteration of the nature of the specific functional interactions (for a $C_{mono}$ of 50μM). Further, to model an unstructured linker, we consider a scenario where the linkers participate in inter-linker interactions alone. Strong inter-linker interactions increased occupied valencies (*Figure 5—figure supplement 1*), for all the concentrations under study, while resulting in much smaller mean largest cluster sizes (*Figure 5A*) during the time-scales accessed by our simulations. Further, the mean radius of gyration ($<R_g^{chain}>$) for the monomers within these clusters shows a sharp transition with an increase in inter-linker interaction strength ($\epsilon_{ns}$). An increase in $\epsilon_{ns}$ from 0.3 kcal/mol to 0.5 kcal/mol results in a sharp decrease in $<R_g^{chain}>$, indicating an onset of linker-driven coil-globule transitions for the polymers within clusters (*Lifshitz et al., 1978*). This effect is also manifest in the increase in the density of the self-assembled clusters upon an increase in $\epsilon_{ns}$. For a $C_{mono}$ of 50μM (*Figure 5B*), we see an increase in cluster density in the range of $\approx 5$ to 50 times of that of the bulk upon variation in linker interactions. A similar trend was observed for a lower $C_{mono}$ of 30μM, with a 10–100 fold variation in degree of enrichment within the cluster upon varying inter-linker interactions (*Figure 5—figure supplement 2*). Therefore, the density transitions could be tuned by varying the free monomer concentrations as well as the properties of the linker. This is consistent with previous equilibrium predictions (*Harmon et al., 2017*; *Choi et al., 2019*) that establish the dependence of density transitions on the properties of intrinsically disordered linkers. The mean density of clusters is also consistent with experimental findings of a $\approx 10$–100 fold enrichment of biomolecules within droplets (*Mitrea and Kriwacki, 2016*; *Li et al., 2012*; *Xing et al., 2018*; *Nott et al., 2015*; *Burke et al., 2015*).

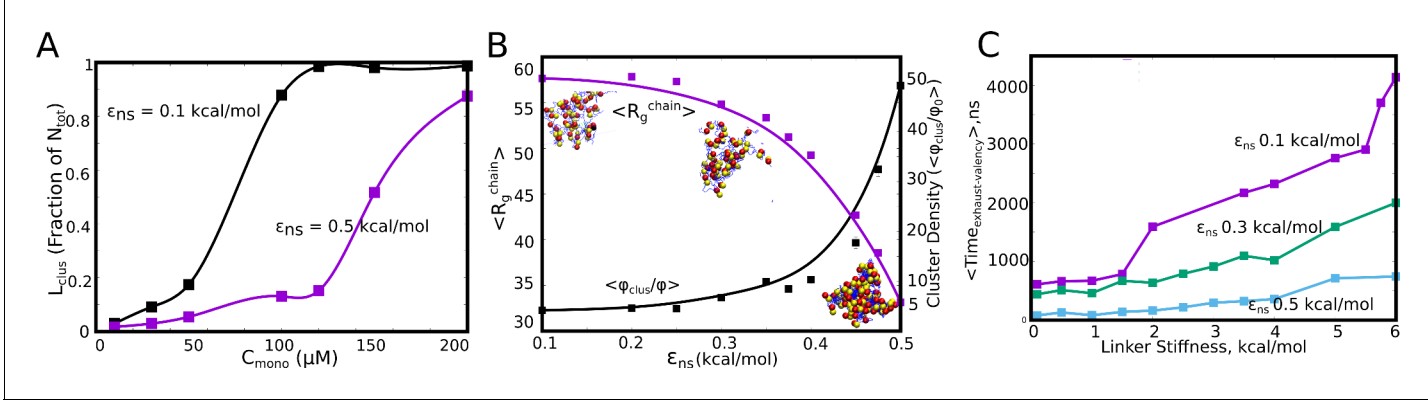

**Figure 5.** Linkers as modulators of self-assembly propensity. (**A**) The size of the largest cluster for flexible linker regions($\kappa$=2 kcal/mol) with varying inter-linker interaction strength (black curve, 0.1 kcal/mol and purple curve, 0.5 kcal/mol). Sticky inter-linker interactions result in smaller cluster sizes. (**B**). Purple Curve – Mean Radius of gyration for the individual-polymer chains ($<R_g^{chain}>$) within a self-assembled cluster as a function of increased inter-linker interactions. Black Curve— Mean density of the clusters as a function of inter-linker interaction strength. (**C**) The mean re-organization times, $T_{exhaust-valency}$ as a function of linker-stiffness, for different values of inter-linker interaction strengths.

The online version of this article includes the following source data and figure supplement(s) for figure 5:

**Source data 1.** Compressed zip file containing the source data for largest cluster size, density of the cluster and radius of gyration of constituent chains, for different values of $\epsilon_{ns}$ plotted in *Figure 5*.

**Figure supplement 1.** Fraction of valencies utilized as a function of increasing inter-linker interaction strength, for (**A**) 10 µM and (**B**) 50 µM free monomer concentration.

**Figure supplement 2.** The probability of finding clusters with varying densities (normalized by the bulk densities) for different values of inter-linker interactions.

**Figure supplement 3.** Comparision between the size distributions of the largest cluster, for flexible ($\kappa$=2 kcal/mol) versus stiff ($\kappa$=5 kcal/mol) linker regions.

The assemblies that ensue at stronger inter-linker attraction are more compact and condensed akin to homopolymer globules (*Lifshitz et al., 1978*). We further probe the manner in which linkers can tune the dynamics of self-assembly, in addition to influencing the equilibrium behavior such as intracluster density. Intramolecular compaction of polymers due to non-specific inter-linker interactions brings specific domains closer in space, leading to a higher likelihood of the exhaustion of specific interaction valencies within small assemblies. In *Figure 5C*, we show how the inter-linker interaction strength can influence the time it takes to exhaust specific interaction valencies within a dimeric cluster ($T_{exhaust-valency}$). For weaker inter-linker attraction ($\epsilon_{ns}$<0.5 kcal/mol), the initial polymer assemblies are less compact (*Figure 5B*) and thereby exhaust valencies within a cluster at a much slower rate (higher $T_{exhaust-valency}$ for dimers with $\epsilon_{ns}$ < 0.5 kcal/mol in *Figure 5C*). An increase in inter-linker interaction propensity results in faster re-organization times for these polymers. The polymers with $\epsilon_{ns}$ = 0.5 kcal/mol exhaust their valencies almost an order of magnitude faster than their 0.1 kcal/mol counterparts (*Figure 5C*). Upon exhaustion of these specific interaction valencies, these clusters can only grow via inter-linker interactions, a phenomenon that could be less dynamic and tunable than the functional interaction driven cluster growth. It must, however, be noted that the observation of further coalescence of clusters formed by sticky inter-linker interactions was limited by the time-scales accessible to the LD simulations. Any alteration to the 'stickiness' of the linker can shift the mechanism of assembly, and thereby result in altered kinetics of cluster growth by modulating the $T_{exhaust-valency}$. Our dimerization simulations show that a second mechanism of slowing down the $T_{exhaust-valency}$ time-scale is by altering the flexibility of the linker region (increasing linker stiffness in *Figure 5C*). Primarily the stiffer linkers lead to more 'open' configurations of spacer regions. This is corroborated by a shift in the cluster size distribution towards larger sizes, for the polymer chains with rigid linkers (*Figure 5—figure supplement 3*). The linker region can thereby serve as a modulator of phase-separation propensity (characterized by the density) and cluster sizes. A variation in the intrinsic properties of the linker, with no modifications to the functional region, can be used as a handle to tune the density of polymers within the condensed phase (*Figure 2* and *Figure 5*). This lends

modularity and functionality to these condensates, with the linker regions influencing both the degree of enrichment and the dynamics of self-assembly.

## Coarse-grained kinetic simulations predict the prevalence of metastable micro-phases at biologically relevant time-scales

The LD simulations help us identify the initial events that mark phase-separation by multivalent polymer chains assembling via finite-valency, specific interactions. However, the model is limited in its ability to access longer, biologically relevant time-scales at which droplets typically form and grow in living cells. We further explore the coalescence of multi-cluster system into a single, large equilibrium-like cluster at longer time-scales that are inaccessible to LD simulations. Here, we employ a coarse-grained approach wherein the whole polymer chain from the bead-spring model (with a fixed valency) is represented as an individual particle on a 2D-lattice. In our simulations, the lattice is populated by $N_{tot}$ such multivalent particles (at varying densities, $\phi_{lattice}$ ) that diffuse freely, at a rate $k_{diff}$, and the number of particle collisions per unit time is proportional to $k_{diff}*\phi_{lattice}$ . In our kMC study, we vary $\phi_{lattice}$ in a range of 0.01 to 0.1 (see Materials and methods section for rationale). When two such particles occupy neighboring sites on the lattice, they interact non-specifically with an interaction strength of $\epsilon_{ns}$ , a parameter that is analogous to the inter-linker interactions in our LD simulations (see *Table 1*). Additionally, two neighboring particles with unfulfilled valencies can form a specific bond (with finite valencies per lattice particle) with a rate of $k_{bond}$. The valency per particle ($\lambda$) here can be utilized to form bonds with one to four potential neighbors, with each pair being part of one, or more than one specific interactions between themselves. However, unlike the irreversible specific interactions in our LD simulations, the specific bonds in the lattice model can break at a rate $k_{break} = k_{bond} *\exp(-\epsilon_{sp})$, where $\epsilon_{sp}$ is the strength of each specific bond. Additionally, clusters can diffuse with a scaled diffusion rate that is inversely proportional to the cluster size ($k_{diff}$ /$S_{clus}$). It must be noted that the time-scale for the first bond formation in the LD simulations is an outcome of two phenomenological rates in this model, $k_{diff}$ and $k_{bond}$. The second time-scale, $T_{exhaust-valency}$, is a time-scale that depends on the $k_{bond}$ and $k_{break}$ parameters in this model. The details of the simulation technique and the various rate processes can be found in the Materials and methods section and described schematically in *Figure 6*.

Using kinetic Monte Carlo simulations (see Materials and methods and *Gillespie, 1977*), we explore the cluster formation (at times reaching a physiologically relevant scale of hours) by varying parameters such as, (a) bulk density of particles on the lattice ($\phi_{lattice}$), (b) rate of bond formation ($k_{bond}$), (c) valency per interacting particle ($\lambda$), and (d) the strength of specific interactions ($\epsilon_{sp}$). With the assumption that, for diffusion-limited self-assembly, the rate of free diffusion $k_{diff}$ is the fundamental time-scale limiting cluster growth, we first explore the relationship between the rate of specific bond formation ($k_{bond}$), and $k_{diff}$ (*Figure 7A* and *Figure 7—figure supplement 1*). It must be noted that the LD simulations employed the assumption that bond formation, upon the two functional domains coming in contact, is an instantaneous event. Here, we show that for values of $k_{bond}/k_{diff} \rightarrow 0$, there is no phase-separation. As the bond formation rate approaches that of free diffusion – corresponding to the instant bond formation assumption in LD simulations – we encountered phase-separated states in our simulations. However, the system largely favors the micro-phase separated state (bluish-red regions in the phase diagram, $L_{clus}$<<1) for low-intermediate $\phi_{lattice}$ even at biologically relevant time-scales of hours in the kMC simulations. In this regime of low-intermediate $\phi_{lattice}$ (bluish-red regions of the kinetic phase diagrams in *Figure 7*), we see denser clusters (*Figure 7—figure supplement 2*) with a wide cluster size distribution (distributions labeled µ1 in *Figure 7—figure supplement 3*). At high values of $\phi_{lattice}$ , we observe a system-spanning macro-phase with lower cluster density (*Figure 7—figure supplement 2*) and a cluster size distribution that favors extremely large clusters co-existing with very small clusters (distributions labeled µ2 in *Figure 7—figure supplement 3*).

Further, this phase diagram (*Figure 7A*) also establishes that for the value of non-specific interaction strength ($\epsilon_{ns}$=0.35 kT, see Materials and methods section for rationale) used here the cluster formation is driven by the finite-valency specific interactions (absence of clustering for $k_{bond}/k_{diff} \rightarrow 0$). For comparison, in *Figure 7—figure supplement 4*, we present the mean cluster sizes for assemblies that are stabilized by non-specific interactions only ($\lambda$=0). The $\epsilon_{ns}$-$\phi_{lattice}$ phase diagram shows that non-specific interaction-driven cluster formation occurs at only high values of $\epsilon_{ns}$ (*Figure 7—*

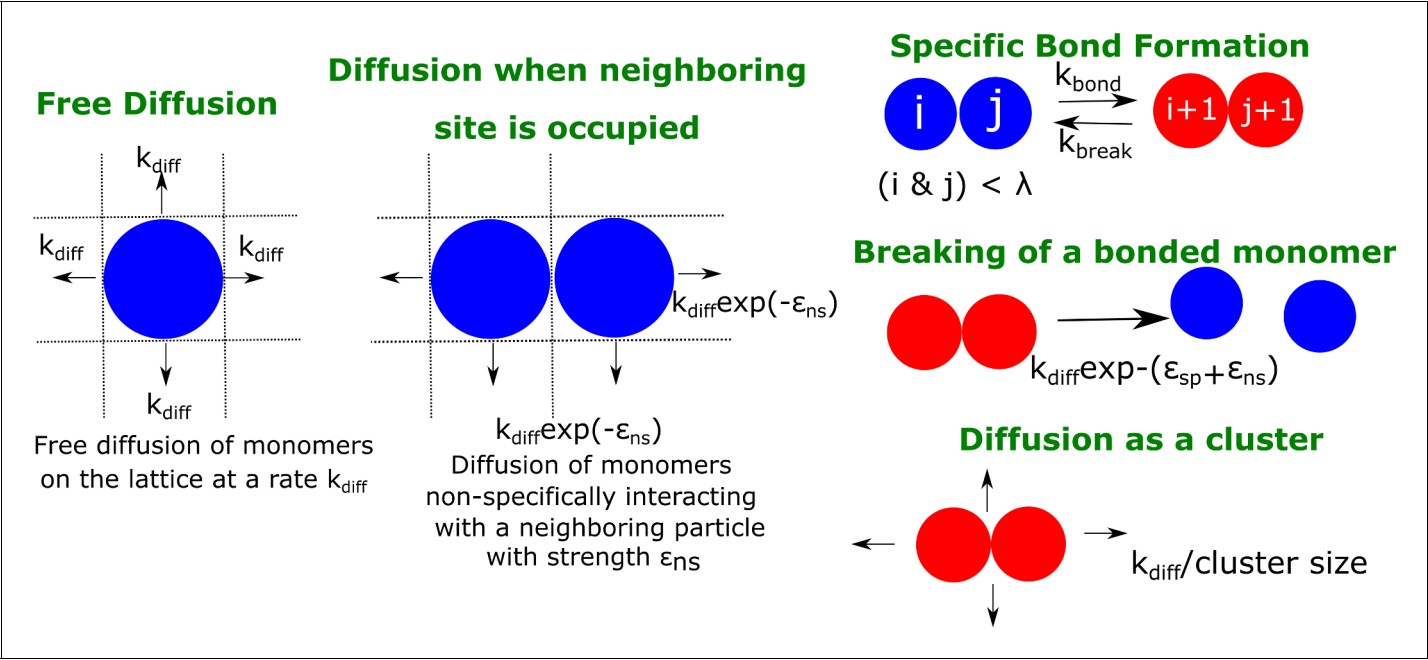

**Figure 6.** A schematic figure detailing the different rates in our phenomenological kinetic model simulated using the Gillespie algorithm. The particles on the lattice can diffuse freely (when there are no neighboring particles) with a rate $k_{diff}$. In the presence of a neighboring particle, a non-specifically interacting monomer can diffuse away with a rate $k_{diff}.exp(-\epsilon_{ns})$. Neighboring particles can also form specific interactions (with fixed valency $\lambda$) at a rate $k_{bond}$ or break an existing interaction with a rate $k_{break}$. Clusters could diffuse at a rate that is scaled by their sizes $S_{clus}$. $\epsilon_{ns}$ and $\epsilon_{sp}$ refer to the strength of non-specific and specific interactions, respectively.

*figure supplement 4B*) . Therefore, cluster formation is contingent on the bonding rate being of the same order as the free-diffusion rate ($k_{bond}$) establishing the validity of the instantaneous bonding assumption in the LD simulations. It is the ratio of $k_{bond}/k_{diff}$ , and not the absolute magnitudes, that is a vital parameter for these simulations. Hence, in all the kMC simulations we set the value of $k_{diff}$ to $1\ s^{-1}$ and vary the ratio $k_{bond}/k_{diff}$ to tune phase separation. It must be noted that, unless mentioned otherwise, the results from the kMC simulations presented here are for a weak non-specific interaction strength of 0.35 kT. All simulations were performed for a time-scale of 2 hr (actual time). As proof of convergence of these simulations, we compare results at the end of 2 hr to those at longer simulation time-scale and show that there is negligible difference in cluster sizes (*Figure 7—figure supplement 5*).

We systematically explored the effect of valency of specific interactions on the extent of phase separation. *Figure 7B* shows kinetic phase diagram with $\lambda$ and $\phi_{lattice}$ as the phase parameters. For smaller $\lambda$ and low $\phi_{lattice}$ , $L_{clus}$ (and $S_{clus}$ ) $<<N_{tot}$ (blue and black regions in *Figure 7B*, and *Figure 7—figure supplements 1* and *3*). This suggests that, at biologically relevant concentrations and timescales, exhaustion of free valencies for particles with smaller $\lambda$ is a crucial determinant of cluster sizes. This phase-diagram is consistent with in vitro experiments involving SH3 and PRM chains with varying valencies, with higher valency molecules displaying a lower critical concentration for phase separation (*Li et al., 2012*).

In addition to the valency of particles, a vital parameter that would determine the droplet sizes is the strength of these specific interactions ($\epsilon_{sp}$ ). As evident from *Figure 7C*, for specific interactions that are extremely weak ($\epsilon_{sp}$<2 kT in *Figure 7C*), there is no significant phase separation. Strikingly, this critical interaction strength (when the largest cluster >= 50% of available monomers) is lower for higher valency particles ($\lambda$=5 curve in *Figure 7C*). Interestingly, the SH3-PRM interaction strength is reported to be in the range of 2–5 kT (*Li et al., 2012*; *Harmon et al., 2017*). A more detailed $\epsilon_{sp}$ - $k_{bond}$ kinetic phase diagram can be found in *Figure 7—figure supplement 6*. The properties of the condensate at biologically relevant time-scales could thus be tuned via different parameters, offering the cell several handles to modulate sizes and morphologies of droplets.

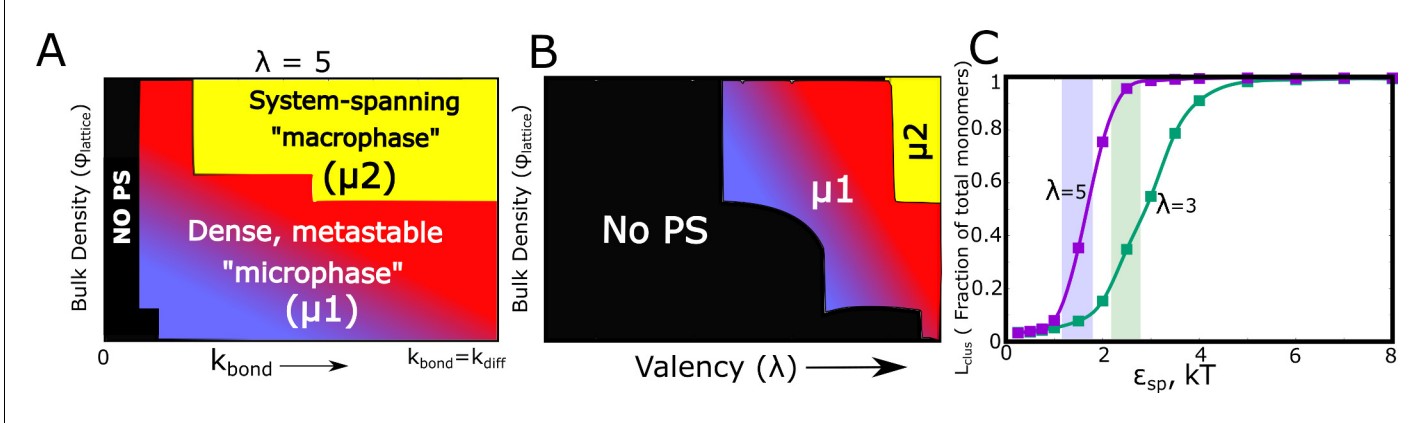

**Figure 7.** Kinetic Monte Carlo Simulations. (**A**) Phase diagram highlighting the different phases (metastable microphase (μ1) or system-spanning macrophase (μ2), and the non-phase separated state (No PS) ) encountered upon increasing $k_{bond}$ (between 0 and $k_{diff}$ ), and the bulk density of monomers ($\phi_{lattice}$ ) within the box. The assembling particles have a valency (λ) of 5 in these simulations. The region shaded yellow represents part of the phase-space where we observe a system-spanning network whereas the bluish-red region represents the metastable micro-phase with a distribution of cluster sizes (see *Figure 7—figure supplement 3* for distributions). As we move from the blue to the red regions in this dynamic phase diagram, the size of the single largest cluster increases (also see *Figure 7—figure supplement 1*). (**B**) Phase diagram highlighting the different phases encountered upon varying valency and bulk density as the phase parameters. The system-spanning macrophase is only encountered for larger valency particles at higher densities. This phase diagram was computed for $k_{bond}=k_{diff}$ and $\epsilon_{ns}$ of 0.35 kT. (**C**) The fraction of monomers in the largest cluster as a function of epsilon, for $k_{diff} = k_{bond}$ and $\phi_{lattice}$ = 0.04. The single largest cluster sizes in all sub-panels of this Figure were computed for a simulation time-scale of 2 hr (with the fundamental timescale of diffusion being set to $k_{diff} = 1\ s^{-1}$ ).

The online version of this article includes the following source data and figure supplement(s) for figure 7:

**Source data 1.** Compressed zip file containing the source data for kinetic phase diagrams in *Figure 7*.
**Figure supplement 1.** Detailed phase diagrams for (**A**) and (**B**) $\phi$-$k_{bond}$, (**C**) $\phi$-λ as the phase parameters.
**Figure supplement 2.** Intra-cluster densities of largest cluster (solid curves) and the corresponding sizes of the single largest cluster (dashed curves) as a function of bulk density, for three different values of λ.
**Figure supplement 3.** Cluster size distributions for varying densities for λ=3 (**A**), and λ=5 (**B**).
**Figure supplement 4.** Inter-protein interaction strengths.
**Figure supplement 5.** Convergence of phase diagrams.
**Figure supplement 6.** Detailed phase diagrams for (**A**) and (**B**) $\epsilon_{sp}$-$k_{bond}$ as the phase parameters.

## Tunability of exchange times for the metastable micro-droplets

In the kMC simulations so far, we discuss the manner in which different phase parameters could shape droplet size distributions. However, the functionality of a condensate hinges not only on the ability of biomolecules to assemble into larger clusters but also to exchange components with the surrounding medium at biologically relevant time-scales. These exchange time-scales are also a measure of the material properties of the droplets themselves (*Guo and Shorter, 2015*; *Xing et al., 2018*; *Feric et al., 2016*). Therefore a systematic understanding of the dependence of molecular exchange times on intrinsic and extrinsic parameters is crucial to get a grasp of the tunability of intracellular self-organization driven by finite-valency specific interactions. In this context, we probed the extent to which the exchange times could be tuned by modulating the intrinsic features of the self-assembling units. Here, we define monomer exchange times as the mean first passage time for a monomer to go from having four neighbors to being completely free. To compute first passage times, we kept track of exchange events from across 100 simulation trajectories of 10 hr each. Our simulations suggest that, for a given valency, a slight increase in interaction strength $\epsilon_{sp}$ within a narrow window could result in a dramatic increase in the size of the clusters (*Figure 7D*). This raises an interesting question—is there an optimal range for these phase parameters that promote phase separation while maintaining the dynamicity of the clusters? In this context, we first computed the mean first passage times for monomer exchange upon systematic variation of $\epsilon_{sp}$ (*Figure 8A*). Interestingly, as with cluster sizes, a slight increase in $\epsilon_{sp}$ results in a dramatic increase in monomer exchange times. For particles with λ=5, a slight increase in $\epsilon_{sp}$ from 2 kT to 2.5 kT, there is a four-fold slow-down in exchange times indicating dramatic malleability in the dynamicity of these assemblies. This

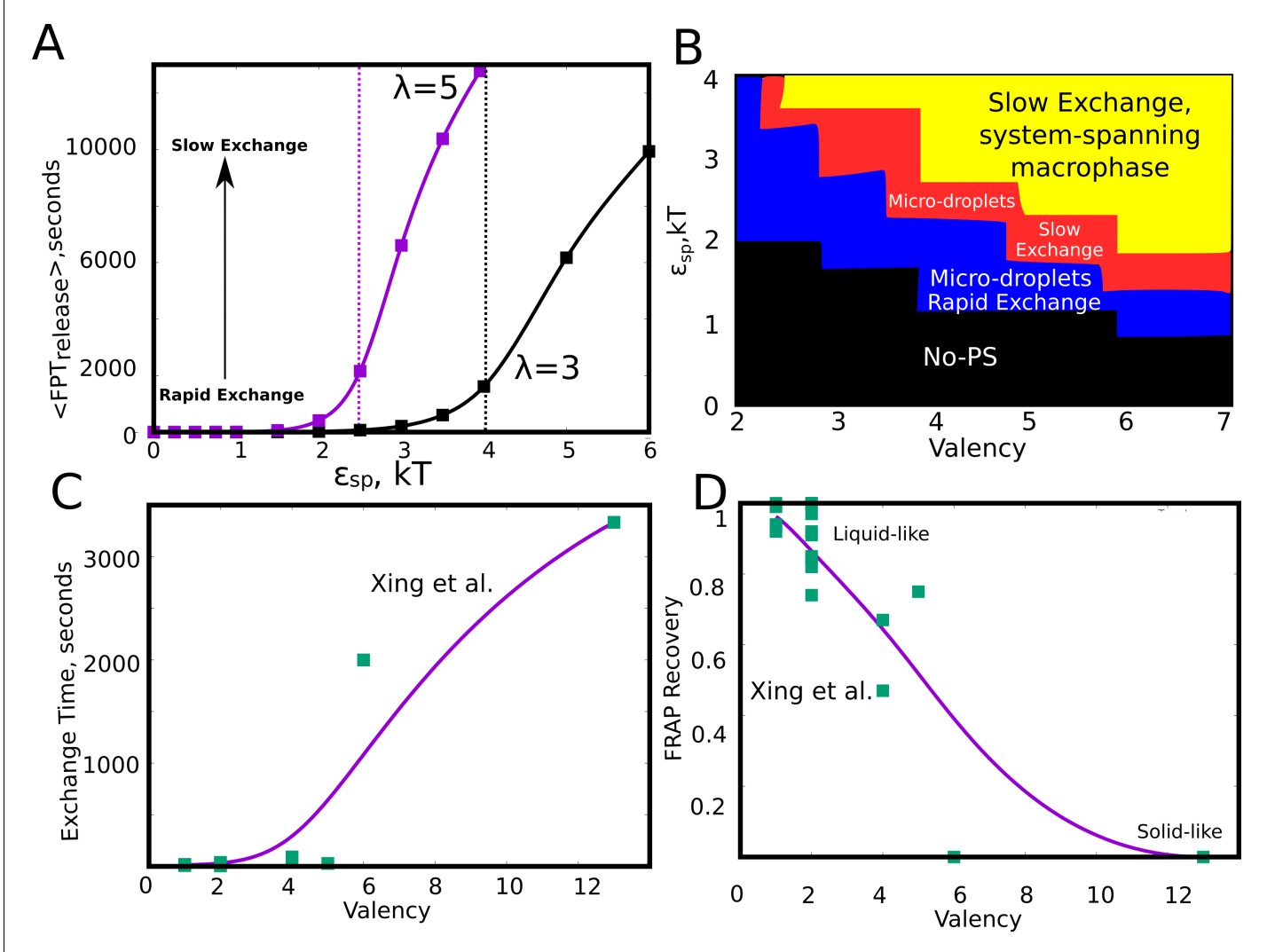

**Figure 8.** Effect of model parameters on the exchange times between monomers and the aggregates. The parameter values used in panels A and B are $\phi_{lattice}$=0.04, $k_{bond}/k_{diff}$ = 1. (**A**) Mean first passage time for the monomers to go from the buried state (with four neighbors) to the free state (with no neighbors) in response to varying values of specific interaction strength. The two curves show the trends for species with different interaction valencies. The dashed solid lines refer to value of $\epsilon_{sp}$ beyond which the largest cluster consumes 50% or more of available monomers. (**B**) The state of the system, for variation in $\epsilon_{sp}$ and $\lambda$ suggests that the system displays remarkable malleability in dynamicity and size distributions. Weak $\epsilon_{sp}$ and a low $\lambda$ results in no phase separation (shaded black). For higher values of both these parameters, the system can access the single large macro-phasic state (shaded yellow), however with a dramatic slowdown in exchange times. For an intermediate range, the system adopts a metastable microphase separated state, with either slow (shaded red) or fast-exchange (shaded blue) dynamics. (**C**) The experimentally determined molecular exchange times for molecules of varying interaction valencies. (**Xing et al., 2018**). (**D**) The extent of recovery after a photobeaching experiment, for interacting species with varying valencies. The data in panel C) and D) has been obtained from a study by **Xing et al., 2018**. The solid curves in C and D are added to guide the eye. The online version of this article includes the following source data and figure supplement(s) for figure 8:

**Source data 1.** Compressed zip file containing the source data for kinetic phase diagrams in **Figure 8**.
**Figure supplement 1.** $<L_{clus}>$ for varying values of $\epsilon_{sp}$ and $\lambda$, for a $\phi_{lattice}$ of 0.04, and a $k_{bond}/k_{diff}$ ratio of 1.

shift from the fast to slow exchange dynamics is less abrupt in case of particles with $\lambda$=3 suggesting that an interplay between $\lambda$ and $\epsilon_{sp}$ could tune the droplets for desired exchange properties. We further varied these two parameters ($\lambda$ and $\epsilon_{sp}$) systematically to probe their effect on cluster sizes (**Figure 8—figure supplement 1A**) and molecular exchange times (**Figure 8—figure supplement 1B**). As expected, for weak $\epsilon_{sp}$ and a low $\lambda$ there is no phase separation (black region in **Figure 8B**). For an intermediate regime in this phase-space, the system is predominantly in a metastable micro-

phase separated state, with either slow (red region in *Figure 8B*) or fast (blue region in *Figure 8B*) molecular-exchange times. However, a system-spanning macro-state is only observed for really large $\lambda$ and $\epsilon_{sp}$, with a dramatic slow-down in exchange times (shaded yellow region in *Figure 8B* and *Figure 8—figure supplement 1B*), suggesting that these assemblies might be biologically non-functional. Valency is, therefore, a key determinant of how frequently a molecule gets exchanged between the droplet and the free medium. Similar observations have been made experimentally by Xing et al. (*Figure 8C and D*) with regards to several condensate proteins featuring different valencies, with low valency species showing exchange times that are orders of magnitude faster than the higher valency ones (*Xing et al., 2018*). Given that functional droplets are tuned for liquid-like behaviour, metastable microphases are, therefore, a likely outcome for dynamically exchanging droplets.

## Effect of solvent viscosity on dynamics of cluster growth

The coarse-grained LD and kMC simulations suggest that kinetic barriers to cluster growth result in the frequently observed state of the long-living multi-droplet state in vitro and in vivo. To highlight this further, we performed LD and kMC simulations where we vary the diffusive properties of the free monomers in solution by tuning solvent viscosity (LD) and diffusion rate (kMC simulations). We first performed LD simulations for solvent viscosities that are 1.5 and 3 times that of water ($\eta=10^{-3}$ Pa .s). As evident from the cluster size distributions at the end of a 20 µs simulation, the

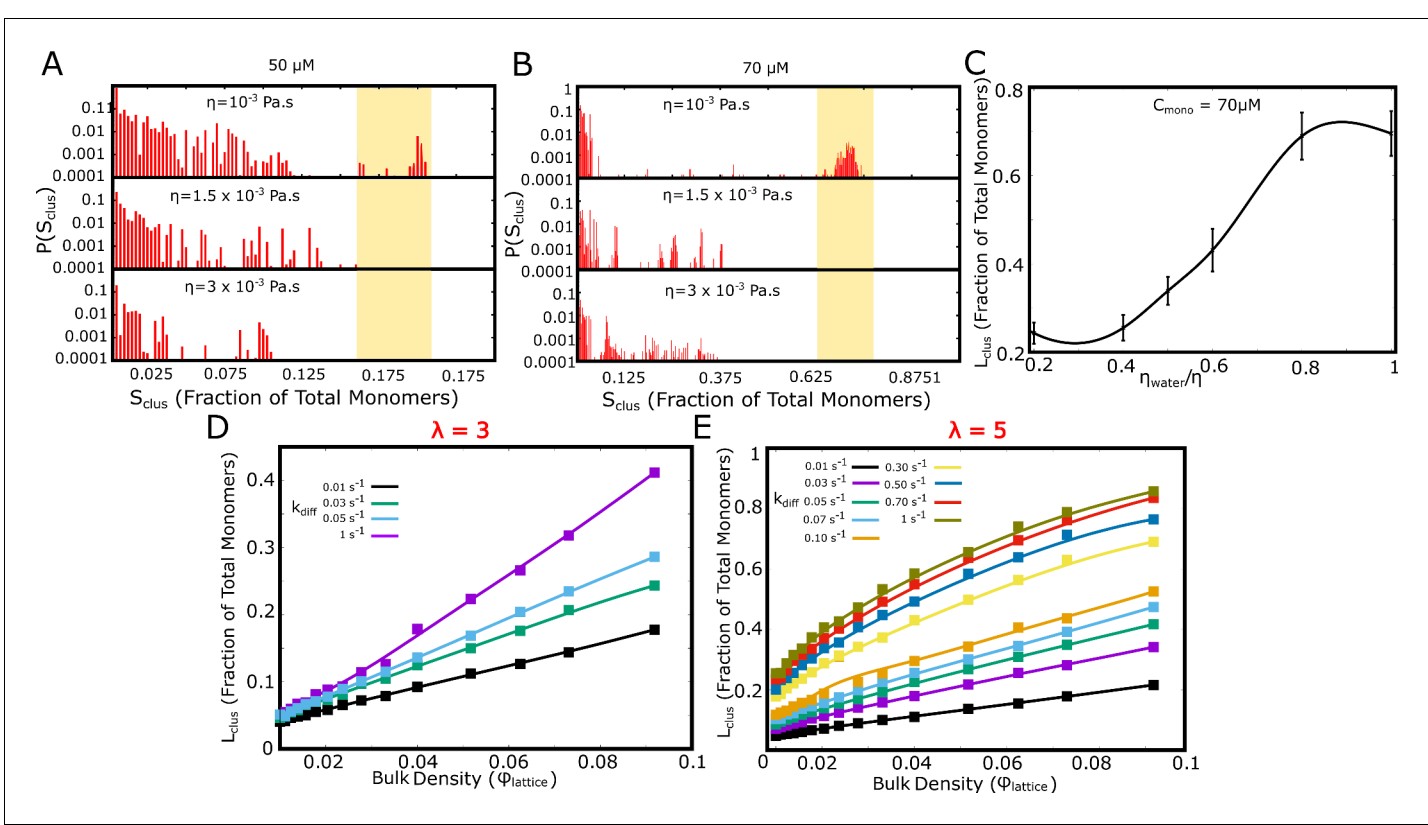

**Figure 9.** Effect of solvent viscosity. Langevin Dynamics Simulations. (**A** and **B**) Cluster size distributions from Langevin dynamics simulations for different values of solvent viscosity and different free monomer concentrations. The shaded region in the two subpanels (**A**) and (**B**) is to highlight that a more viscous solvent results in larger clusters disappearing from the distribution. (**C**) The size of the single largest cluster as a function of solvent viscosity for a free monomer concentration of 70 µM. (**D** and **E**) Monte Carlo Simulations. The size of the single largest cluster as a function of density for varying rates of diffusion, for specific interaction valencies $\lambda=3$ (**D**) and $\lambda = 5$ (**E**).

The online version of this article includes the following source data and figure supplement(s) for figure 9:

**Source data 1.** Compressed zip file containing the source data for mean largest cluster sizes (and size distributions) from LD and MC simulations studying the effect of solvent viscosity.

**Figure supplement 1.** $L_{clus}$ as a function of free monomer concentrations in LD simulations performed with various solvent viscosities.

larger clusters were not observed for more viscous solvents (shaded region in *Figure 9A and B*). This is further corroborated by the smaller cluster sizes at higher solvent viscosities (*Figure 9C*). This suggests that slowing down the diffusion-limited time-scale for higher viscosities could result in a further reduction in the observable cluster sizes at finite time-scales relevant to biology. At higher viscosities, we observe large clusters only at higher free monomer concentrations (*Figure 9—figure supplement 1*). We further probed the role of diffusion-limited time-scale on the observed cluster sizes at a fixed time-scale of 2 hr in our kinetic Monte Carlo simulations. As with our LD simulations, slower diffusion rates result in smaller cluster sizes for a simulation time-scale of 2 hr, both for a $\lambda$ of 3 and 5 (*Figure 9D and E*). These results are consistent with previous in vitro studies which suggest that solvent viscosity could influence the rate of droplet coalescence (*Kaur et al., 2019*). These results make a key testable prediction of this work pointing to the vital role of dynamics in the formation of membraneless organelles as metastable micro-droplets positing that the measurable quantities such as droplet sizes depend on dynamic quantities of the solvent such as viscosity. If micro-droplets were formed at equilibrium, such dependencies would not be observed (*Harmon et al., 2017*; *Brangwynne et al., 2015*). While initial indications point out to the crucial role of solvent viscosity (*Kaur et al., 2019*) more experimental studies are needed to confirm or refute this key prediction of our study.

## Discussion

### Long-living metastable droplets: a potential signature of multivalent heteropolymers

Membrane-less organelles are heterogeneous pools of biomolecules which localize a high density of proteins and nucleic acids (*Banani et al., 2017*; *Feric et al., 2016*; *Boeynaems et al., 2019*). An interesting feature of several complex macromolecules (*Mitrea and Kriwacki, 2016*; *Li et al., 2012*; *Harmon et al., 2017*) that constitute these droplets is 'multivalency' stemming from multiple repeats of adhesive domains (*Banani et al., 2017*; *Li et al., 2012*; *Patel et al., 2015*). These adhesive domains can bind to complementary domains on other chains, thereby facilitating phase separation. In this study, we model this phenomenon as that of self-associative polymers that possess folded domains (represented as idealized spheres in *Figure 1*) separated by flexible linker regions. Recent computational studies have employed similar models to characterize the equilibrium state of these polymer systems, notably the coarse-grained simulations by *Harmon et al., 2017* and *Choi et al., 2019*. These studies employ the 'sticker and spacer' model to understand the phase behaviour of linear multivalent polymers (*Harmon et al., 2017*), mainly focusing on the nature of the phase-separated state at equilibrium. Using lattice-polymer Monte Carlo simulations, they establish the role of intrinsically disordered linker regions as molecular determinants that dictate the equilibrium state of a system of associative polymers that interact via non-covalent interactions (*Harmon et al., 2017*). Crucially, these works focus on the cross-linked gel-like nature of the equilibrium state of these polymers and establish the underlying thermodynamic landscape governing phase separation of spacer-sticker polymers. The observation of the single, large equilibrium-like droplet phase in vitro establishes the robustness of the existing thermodynamic understanding of the phenomenon. However, experiments also report the presence of multiple, co-existing droplets (with equilibrium-like droplet micro-environments) that remain stable at biologically relevant time-scales (*Brangwynne et al., 2011*; *Kilchert et al., 2010*; *Dine et al., 2018*; *Wegmann et al., 2018*) in vivo and in vitro. Minimal models by *Wurtz and Lee, 2018b* point towards the potential role of ATP-dependent processes (*Weber et al., 2019*) in suppressing Ostwald ripening, and thereby the stabilization of the multi-droplet system. However, the potential role of a more general, ATP-independent mechanism in establishing a long-living multi-droplet phase has not yet been explored.

The deviation from the equilibrium picture raises a number of interesting questions –how does the multivalent, multi-domain architecture of proteins influence the suppression of droplet growth into a single macroscopic phase by coalescence? Does the physics of associative polymers such as multivalent proteins differ from those of simple, homopolymer chains? Also, in the context of the size-dependent function of membrane-less organelles, can the intrinsic features of the self-assembling chain act as handles that control the dynamics and size distributions of the droplets? In this paper, we argue that, even in the absence of active processes, the finite interaction valencies for the

sticker-spacer architecture could suppress the coalescence of multiple metastable droplets into a single large droplet.

## Exhaustion of specific interaction valencies is a root cause of arrested LLPS even in the absence of active processes

First, we studied self-assembly by multivalent polymers whose adhesive domains interact via stable, 'non-transient' interactions to understand the early events in the growth process. We observed that, except for extremely high concentrations where polymers form a system spanning network (*Figure 2A*), the most feasible scenario at smaller concentrations is that of co-existing clusters with a concentration-dependent distribution of cluster sizes. While the non-monotonic dependence of density-transitions on concentration is consistent with equilibrium predictions (*Harmon et al., 2017*), the prevalence of multiple clusters with a distribution of cluster sizes at lower $C_{mono}$ is a key deviation. Crucially, in the parameter regime where we observe dense clusters (*Figure 2*), the available interaction valencies get consumed within smaller-sized assemblies (*Figure 3*), making them inert for further cluster growth. Our results suggest that two critical time-scales decide whether a cluster continues to grow further – a) concentration-dependent time-scale of two chains (or clusters) encountering each other and forming the initial functional interaction, and, b) the exhaustion of valencies within a small cluster, a time-scale dependent on intrinsic features of the polymer (*Figure 4*). Crucially, these two time-scales are sensitive to subtle modifications in the self-assembling polymer chain (*Figure 5*). Therefore, while the underlying equilibrium landscape drives the process of self-assembly, the metastable multi-droplet system could be a dynamic outcome that is characteristic of polymers with multiple stickers. Our findings are consistent with the predictions from the 'sticky reptation' theory proposed by Semenov and Rubinstein that suggests that the dynamics of associative polymers with multiple stickers exhibits a slow-down for high degrees of association (*Rubinstein and Semenov, 2001*). At higher degrees of association for multivalent polymers, with fewer free sites, the effective lifetime of each interaction becomes higher, making the recruitment of a new chain progressively less likely (*Rubinstein and Semenov, 2001*; *Leibler et al., 1991*). The role of physical crosslinks has also been reported to play a key role in tuning the viscoelasticity of protein assemblies (*Roberts et al., 2018*).

Modifications to the linker region can also result in altered densities of the self-assembled state, with a 10–100 fold enrichment in molecular concentrations within the clusters (*Figure 5* and *Figure 5—figure supplement 3*), consistent with the experimentally reported degrees of enrichment within condensates (*Nott et al., 2015*; *Mitrea and Kriwacki, 2016*; *Li et al., 2012*) and previous theoretical studies showing the role of the linker in modulating phase separation (*Harmon et al., 2017*). Using LD simulations involving reversible and irreversible functional interactions, we establish a potential physical mechanism determining microphase separation in membrane-less organelles, with the finite nature of the specific interactions driving the peculiar phenomenon. Therefore, the highly cross-linked nature of the phase-separated state (*Harmon et al., 2017*; *Rubinstein and Semenov, 2001*) would result in the early droplets not being able to transition into a single large droplet even for reversible, transient interactions.

## Bridging the gap between the early and biologically relevant timescales

To reach the time scales relevant to biology, we employed a coarse-grained kinetic model where each polymer (from the LD simulations) is represented as a diffusing reaction centre on a 2D lattice, which can interact either non-specifically (mimicking inter-linker interactions) or specifically with neighboring centers. The difference between the two types of interactions is that the number of specific interactions that each centre can make is limited by its valency. In an extension of the LD model, specific interactions are stable yet reversible and can form and break with rates dictated by detailed balance. In contrast to the 2D-lattice aggregation models employed by *Dine et al., 2018*, our model incorporates interaction valencies explicitly. Our kMC simulations also incorporate diffusion of entire clusters to allow coalescence, unlike the Dine model where clusters can grow only by monomer diffusion events and Ostwald ripening – an extremely slow process. Consistent with our LD simulations, our kinetic Monte-Carlo simulations of the phenomenological model reveal that for time-scales relevant to biology, fully percolated, system-spanning clusters ($L_{clus} \longrightarrow N_{tot}$) are observed only for high bulk density ($\phi_{lattice}$) of monomers (*Figure 7*). Further, the phenomenon of exhausted adhesive

valencies is more prominent for species with lower valency (fewer adhesive domains in the prototypical polymer), as evident from much smaller sizes of the largest cluster after an hour-long (real time) simulation run. The sensitivity of the cluster sizes to these phase parameters (valency, bulk density, interaction strengths) suggests that sizes and morphology of membrane-less organelles are amenable to tight control. In the context of the size-dependent function of membrane-less organelles, this tunability becomes critical. This is consistent with previously proposed mechanisms of organelle growth control by limiting the availability of components critical for droplet growth (*Goehring and Hyman, 2012*).

The kinetically trapped nature of the multi-droplet phase was further confirmed by the observation that cluster sizes (for the same simulation time) decrease upon slowing down the diffusion-limited time-scale by varying solvent viscosity (LD) or phenomenological diffusion coefficient (kMC) (*Figure 9*). The lattice-diffusion model, despite its minimalistic nature, captures the experimentally established relationship between molecular valency and critical concentration (*Li et al., 2012*) for phase separation (*Figure 7B*). An interplay between the valency of the generalized polymer and the strength of interactions can also alter the exchange times of molecules with the bulk medium dramatically. A slight shift in either these valencies or interaction strengths could result in a change in exchange rates by orders of magnitude. Further, for regions of the parameter space ($\epsilon_{sp}$ and $\lambda$) that favor large clusters ($S_{clus}$ —> $N_{tot}$) separation, we observe a dramatic slow-down in molecular exchange times. In other words, for parameters that result in a fast-exchanging condensed state, the metastable microphase is the most favored outcome (*Figure 8B* and *Figure 8—figure supplement 1*). Such a discontinuity makes these systems extremely sensitive to mutations (*Wang et al., 2018*) that might cause a shift in dynamics and eventual loss of function of these droplets. Also, such shifts could also make these systems extremely responsive to non-equilibrium processes such as RNA-processing, side-chain modifications (acetylation, methylation) that are often attributed to modulating condensate dynamics (*Wang et al., 2018*; *Hofweber and Dormann, 2019*; *Wang et al., 2014*). The differential exchange times in response to variation of interaction parameters in our model lend further support to the scaffold-client model (*Xing et al., 2018*). The scaffolds that are slower exchange species with higher valencies acting could, therefore, recruit faster exchange clients with lower valencies. The valencies and strength of interactions could have thus evolved to achieve exchange times that ensure the functionality of spatial segregation via liquid-liquid phase separation.

A multi-domain sticker-spacer architecture allows for separation of two functions with the folded domains (conserved) performing functional role while the spacer regions being modified over time to tune the propensity to phase separate and also the material nature of the condensate. Overall, our multi-scale study shows that the block co-polymer-like organization of these multivalent proteins, with finite specific interactions driving phase separation, could manifest itself in the metastable multi-droplet state (*Figure 10*). A switch in the driving force for self-assembly, from the specific to non-specific interactions via sticky linkers (*Wang et al., 2018*), could alter not only the kinetics of assembly but also have implications in disorders associated with aberrant phase separation (*Ramaswami et al., 2013*; *Patel et al., 2015*). Our study paves the way towards the rational design of phase-separating polymers which is of vital importance for addressing their role in disease and function.

## Testable predictions

The emphasis of the current study is on the metastable nature of the long-living micro-droplet-like state of multivalent biopolymers. A signature of metastability would be the dependence of the observable quantities such as droplet sizes and size distribution on the dynamics of self-assembly process – a phenomenon not observed at equilibrium. Here, we propose a set of potential experiments that could be performed to test our key prediction of the dynamic origin of the multi-droplet system.

- Our LD and kMC simulations suggest that the observed droplet size distributions (for a fixed biologically relevant time-scale) would depend on the viscosity of the solvent. At higher viscosities, the slowing down of droplet coalescence would therefore result in a shift in the droplet size distribution towards smaller sizes. A systematic experimental study of phase separation (at a time-scale of hours) in a range of solvent viscosities would provide an essential test of our findings.

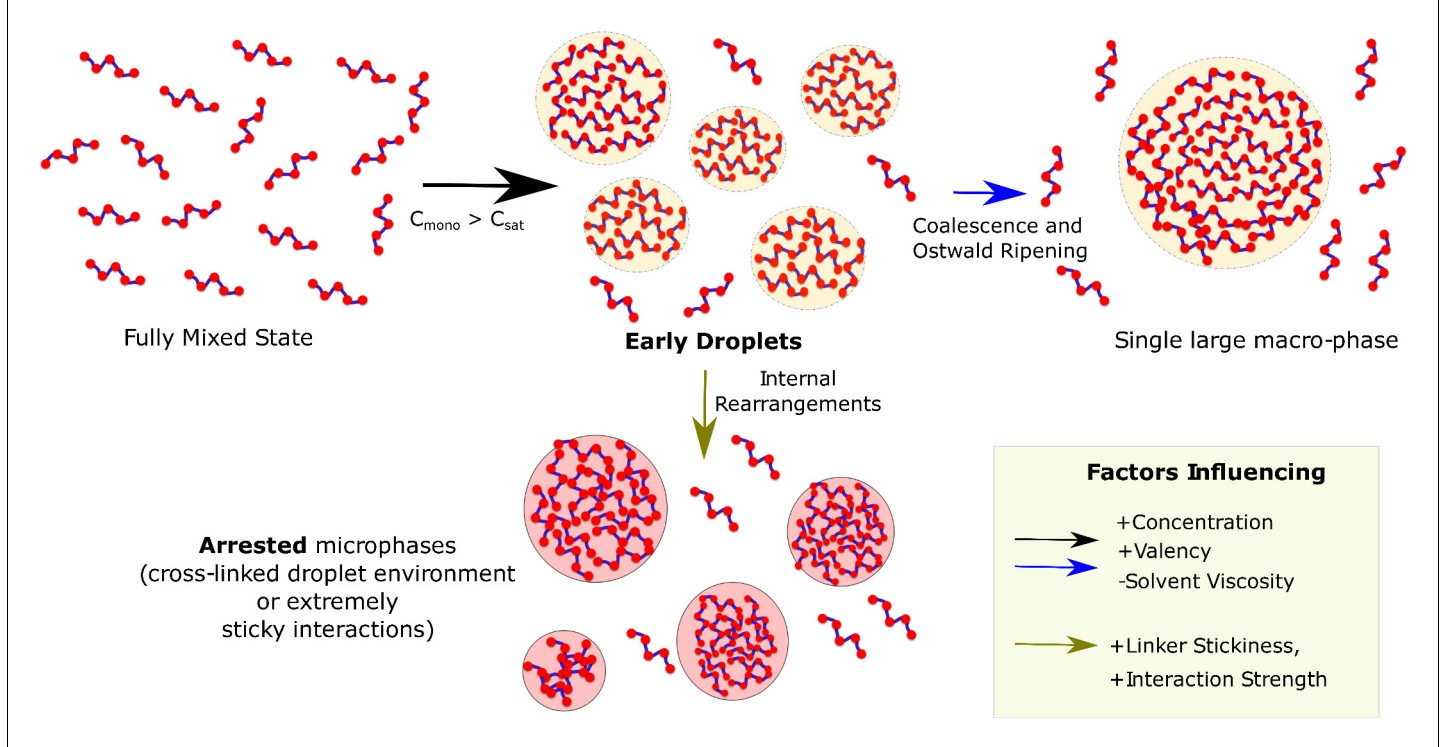

**Figure 10.** Graphical Summary. Muli-valent proteins can exist in two Flory-like equilibrium states, the fully mixed solvated state and a single large macro-phase separated state (when free monomer concentration ($C_{mono}$) exceeds the critical concentration ($C_{sat}$)). Above the critical concentration, the proteins first phase separate into multiple droplet phases (black arrow) which are long-living, metastable structures. However, the growth of these individual droplets to a single large macro-droplet (blue arrow) through coalescence and Ostwald ripening is unlikely at biologically relevant timescales. The multiple metastable micro-droplets can also age into highly cross-linked structures (green arrow) that saturate interactions within themselves and thus cannot grow further. Extremely stable functional interactions or sticky inter-linker interactions could also result in these arrested microphases.

- Our phenomenological kinetic model predicts that the dynamic, droplet phase exists in a very narrow parameter regime (concentration, valency, interaction strength) and that a subtle change in parameters could result in either a fully mixed-phase or clusters that exhibit a dramatic slowdown in exchange times. A more systematic experimental study of droplet dynamics as a function of interaction valencies would be required to establish the veracity of these predictions. Engineered multivalent proteins such as the SH3-PRM system (*Li et al., 2012*) could serve as useful model systems for such a study.
- Another key observation of this model is the ability of the linker region to modulate the nature of phase separation. Previously, synthetic multivalent protein model systems have been used to study LLPS in vitro (*Li et al., 2012*). Given that our dynamic model predicts small, condensed aggregates for strong inter-linker interactions, systematic studies of droplet size distributions in a range of inter-linker interaction strengths would establish how the observable quantities are sensitive to intra- and inter-cluster dynamics.

## Materials and methods

### Langevin Dynamics simulations
#### Force field
The polymer chains in the box are modelled using the following interactions. Adjacent beads on the polymer chain are connected via harmonic springs through the potential function,

$$E_{stretching} = k_s \sum_{i=1}^{M-1} (|\vec{r_i} - \vec{r_{i+1}}| - r_0)^2, \tag{1}$$

$\vec{r_i}$ and $\vec{r_{i+1}}$ refer to the adjacent $i^{th}$ and $(i+1)^{th}$ bead positions, respectively. Here, $r_0$ is the equilibrium bond length and $k_s$ represents the spring constant. This interaction ensures the connectivity between the beads of a polymer chain.

This spring constant, $k_s$, in our LD simulations is set to 5 kT/Å$^2$, in order to ensure rigid bonds between neighboring beads of the same chain. The equilibrium length $r_0$, for bonds connecting adjacent linker beads, is set to 4.5 Å while the same for specific interactions between functional domains is set at 20 Å.

To model semi-flexibility of the polymer chain, any two neighboring bonds within the linker regions of the polymer chains interact via a simple cosine bending potential

$$E_{bending} = \kappa \sum_{i=1}^{M-2} (1 - cos\theta_i), \qquad (2)$$

where $\theta_i$ describes the angle between $i^{th}$ and $(i+1)^{th}$ bond while $\kappa$ is the energetic cost for bending (bending stiffness of linker). We vary the bending energy parameter ($\kappa$) to model rigid ($\kappa$>2 kcal/mol) or flexible linkers ($\kappa$=2 kcal/mol).

The non-bonded isotropic interactions between linker beads and linker-functional interactions were modelled using the Lennard-Jones (LJ) potential,

$$E_{nb} = 4\epsilon_{ns} \sum_{i<j} \left[ \left( \frac{\sigma}{|\vec{r_i} - \vec{r_j}|} \right)^{12} - \left( \frac{\sigma}{|\vec{r_i} - \vec{r_j}|} \right)^{6} \right], \qquad (3)$$

for all $|\vec{r_i} - \vec{r_j}| < r_c$, where $r_c$ refers to the cutoff distance beyond which the non-bonded potentials are neglected. The LJ potentials were truncated at a distance cutoff of 2.5σ. σ was set to 4.2 Å (roughly that of amino acid) for linker beads and 20 Å for functional domains.

The strength of the attractive component of this potential, $\epsilon_{ns}$, was varied to achieve varying degrees of inter-linker interactions in our simulations. In our simulations, the linker regions participate in inter-linker interactions only. The strength of this inter-linker $\epsilon_{ns}$ in our simulations was varied in the range of typical strengths of short-range interactions in biomolecules. We vary the $\epsilon_{ns}$ for pairwise inter-linker interaction (per pair) in the range of 0.1 kcal/mol ($\approx$0.2 kT) to 0.5 kcal/mol ($\approx$1 kT) (*Sheu et al., 2003*). For comparison, strong non-covalent interactions such as the H-bonds are known to be of the order of 0.5 kcal/mol to 1.5 kcal/mol in solvated proteins. Similar values of isotropic, short-range interactions have been used in conventional coarse-grained protein force-fields such as the MARTINI (*Marrink et al., 2007*; *Monticelli et al., 2008*).

## Modeling specific interactions

The specificity of the functional interactions is modelled by imposing a valency of 1 between different complementary specific bead types (red and yellow beads in *Figure 1*) via a bonding vector attached to each bead. Valency of 1 per site means that each specific interaction site can only be involved in one such interaction. The total valency of an individual polymer chain ($\lambda$) is the number of adhesive sites that belong to a single chain. Bond formation (modelled using stochastically forming harmonic springs) occurs with a probability ($P_{form}$) if two complementary beads are within a defined interaction cutoff distance ($r_c$). The spring constant for functional interactions is set at 2 kT/Å$^2$. In simulations with irreversible functional interactions, a bond, once formed, remains stable for the duration of the simulations. In order to implement reversible specific interactions, we introduce a bond breakage probability ($P_{break}$) when the inter-domain distance stretches beyond a breakage cutoff – $r_{break}$. In our reversible specific interaction simulations, a bond breaks with a probability ($P_{break}$) of 1 when the two domains move 2.2 Å from the equilibrium distance of the spring (used to model the specific interaction). The strength of the spring at the breakage point is $\approx$4 kT, a value that is within the range of interaction strengths previously probed in lattice simulations of sticker-spacer polymers (*Harmon et al., 2017*).

## LD simulations details

The dynamics of these coarse-grained polymers was simulated using the LAMMPS molecular dynamics package (*Plimpton, 1995*). In these simulations, the simulator solves for the Newtons's equations of motion in presence of a viscous drag (modeling effect of the solvent) and a Langevin thermostat

(modeling random collisions by solvent particles) (*Schneider and Stoll, 1978*). The simulations were performed under the NVT ensemble, at a a temperature of 310 K. The mass of linker beads was set to 110 Da while the mass of the idealized functional domains (red and yellow beads in *Figure 1*) was set to 7000 Da that is approximately equal to the mass of the SH3 domain. The size of the linker beads was set at 4.2 Å (of the same order as amino acids) while that of the functional domains was set at 20 Å ($\approx$ size of a folded SH3 domain *Musacchio et al., 1992*). The viscous drag was implemented via the damping coefficient, $\gamma = m/6\pi\eta a$. Here, m is the mass of an individual bead, '$\eta$' is the dynamic viscosity of water and 'a' is the size of the bead. An integration time step of 15 fs was used in our simulations, and the viscosity of the surrounding medium was set at the viscosity of water ($10^{-3}$ Pa.s). Similar values of these parameters have been previously employed for coarse-grained Langevin dynamics simulations of proteins (*Bellesia and Shea, 2009*).

## Kinetic Monte carlo simulations

To assess biologically relevant time-scales, we develop a phenomenological kinetic model wherein the individual multivalent polymer chains are modelled as diffusing particles with fixed valencies (*Figure 6*). In order to study the temporal evolution of a system of diffusing particles on a lattice, we employ the (*Gillespie, 1977*) or the kinetic Monte Carlo simulation, a technique used for modelling a system of chemical reactions. This algorithm has previously been used to model biological processes as diverse as gene regulation (*Parmar et al., 2014*) and cytoskeletal filament growth (*Ranjith et al., 2009*).

### Reactions (processes), and interactions modelled in the simulation

We begin the simulation with a set of $N_{tot}$ particles randomly placed on a 2D square lattice. The density of particles on the lattice ($\phi_{lattice}$) is a measure of the bulk initial concentration of monomers in these simulations. Each particle in our lattice Monte carlo simulations is a coarse-grained representation of the bead-spring polymer chains in the LD simulations, with an effective valency ($\lambda$) that is a simulation variable. Assuming that there are unoccupied neighboring sites, the monomers can diffuse on the lattice in any of the four directions (in 2D) with a rate $k_{diff}$. Particles occupying adjacent sites on the lattice experience a weak, non-specific interaction (of strength $\epsilon_{ns}$) that is isotropic in nature. This attractive isotropic force is analogous to the inter-linker interactions in the LD simulations. A functional bond (specific interaction) can stochastically form between any pair of neighboring particles, provided both particles possess unsatisfied valencies. The rate of bond formation between a pair of particles with unsatisfied valencies is $k_{bond}$. On the other hand, an existing functional bond can break with a rate $k_{break}$ that is equal to $k_{bond}.\exp(-\epsilon_{sp})$ in magnitude. $\epsilon_{sp}$ is the strength of each functional interaction. Additionally, the entire cluster that any given monomer is a part of can diffuse in either of the four directions with a scaled diffusion rate that is inversely proportional to the size of the cluster ($k_{diff}/S_{clus}$). A particle that is part of a cluster can diffuse away from the cluster with a rate $k_{diff}.\exp(-\sum \epsilon_{sp} + \epsilon_{ns})$. Here, $\sum \epsilon_{sp} + \epsilon_{ns}$ is the magnitude of net interactions that any particle is involved in. These rates are schematically described in *Figure 6*.

### Details of the Gillespie simulation algorithm

At every time-step of the simulation, these rates – diffusion of monomer, diffusion of the cluster that a monomer is a part of, rate of bond formation and rate of bond breakage – are initialized for every monomer based on the current configuration of the system. Note that, for any given configuration, all or only a subset of these events could be possible. We then advance the state of the system by executing one reaction at a time. The probability of each event is computed, and it is equal to $r_i/\sum\sum r_i^j$. Here i and j refer to the identity of the monomer and the event type (diffusion, cluster diffusion, bond formation), respectively. The denominator in this term is the sum of the rates of all the possible events for a given configuration (for all monomers), referred to as $r_{\text{total}}$. At any given time-step, given the set of probabilities for each event, we choose the event to be executed by drawing a uniformly distributed random number $z_1$ (between 0 and 1). The configuration of the system is then updated based on the event that gets chosen at any time-step. The Gillespie Exact Stochastic Simulation technique (*Gillespie, 1977*) is a variable time-step simulation approach wherein the simulation time is advanced using the following expression, $\delta t = -(1/r_{total})ln(z_2)$, where $z_2$ is a uniform random number. Here, $r_{total}$ is the sum of the rates of all possible events at a given

simulation time-step. This $\delta t$ is based on the assumption that the waiting times between any two events are exponentially distributed. The above algorithm has to be iterated several times such that each reaction has been fired multiple times, suggesting the system has reached steady-state behaviour. A detailed account of the algorithm can be found in the seminal work by *Gillespie, 1977*.

### Computing cluster sizes in kMC and LD simulations

Cluster size calculations form the basis of the current study. In order to compute cluster sizes in LD simulations, we used the gmx-clustsize module within the GROMACS molecular dynamics package (*Van Der Spoel et al., 2005*). When the closest distance between any two functional domains from different polymer chains is within a 22.5 Å radius (interaction radius for functional domains in the simulation), we consider them to be part of the same cluster. We use the snapshots from the final 1 μs of the trajectory (and from five independent trajectories) to perform cluster size computations. In order to compute cluster sizes and distributions in the kMC simulations, we employed the Hoshen-Kopelman algorithm (*Hoshen and Kopelman, 1976*), which is routinely used for studying percolation problems.

### Rationale behind phase parameters in KMC simulations

Two key phase parameters in our kMC simulations are valency per interacting particle and bulk density of particles on the lattice. We performed simulations for valencies ranging from 3 to 6, typical of multivalent proteins forming membrane-less organelles. Unlike non-specific interactions which can be only one per neighbor, thereby a maximum of 4 per particle on a 2D-lattice, there could be more than one specific interactions between a pair of neighboring particles, as long as both participating members have unsatisfied valencies. This multiplicity of specific interactions between nearest neighbors in this model reflects the fact that each polymer contains $\lambda > 1$ globular domain that can engage in specific interactions with globular domains from neighboring protein. The bulk density of particles is defined as, $\phi_{lattice} = N_{tot}/L^2$, where $N_{tot}$ and $L$ are the number of particles and lattice size, respectively). In our kMC simulations, we varied $\phi_{lattice}$ within the range of 0.01 to 0.1, a range that is consistent with the analogous parameter for LD simulations, the occupied volume fractions within the LD simulation box. For instance, a free monomer concentration ($C_{mono}$) of 10 μM corresponds to a volume fraction of 0.008, which increases to 0.17 for 200 μM. Further, the phase diagrams for KMC simulations were primarily computed for a weak non-specific $\epsilon_{ns}$ of 0.35 kT, chosen in order to focus on specific interaction-driven cluster growth. In the kMC simulations, $\epsilon_{ns}$ refers to the net non-specific interaction for a pair of monomers as opposed to a pair of interacting linker beads in the LD simulations. It must, however, be noted that the interaction strength for a pair of interacting chains is not merely additive (with the length of the polymer chain). A potential explanation for this could be found in earlier studies wherein it has been argued that smaller segments ($\approx$ length of 7–10 amino acids) within a long, solvated polymer chain (like IDPs) could behave like independent units referred to as blobs (*Pappu et al., 2008*). Since the degree of coarse-graining employed in kMC is of the order of one particle per polymer chain, it lacks the microscopic degrees of freedom of the bead-spring polymer chain (in LD simulations). Hence we do not employ a higher non-specific interaction strength in our phase plot computations. As a proof of principle, we show the mean pairwise interaction energies for dimers from the LD simulations, for a weak inter-linker interaction strength of 0.1 kcal/mol (*Figure 7—figure supplement 4A*) and a corresponding phase-diagram for non-specifically driven interactions (*Figure 7—figure supplement 4B*). The mean cluster sizes and molecular exchange times in the kMC simulations were computed over 100 independent kMC trajectories.

## Acknowledgements

This work was supported by NIH RO1 GM068670. We are grateful to Alexander Grosberg, Mark Miller and Ranjith Padinhateeri for useful discussions.

## Additional information

### Funding

| Funder | Grant reference number | Author |
|---|---|---|
| National Institutes of Health | RO1 GM068670 | Eugene I Shakhnovich |

The funders had no role in study design, data collection and interpretation, or the decision to submit the work for publication.

### Author contributions

Srivastav Ranganathan, Conceptualization, Validation, Investigation, Visualization, Methodology, Writing - original draft; Eugene I Shakhnovich, Conceptualization, Formal analysis, Supervision, Funding acquisition, Validation, Investigation, Visualization, Methodology, Writing - original draft

### Author ORCIDs

Eugene I Shakhnovich https://orcid.org/0000-0002-4769-2265

### Decision letter and Author response

Decision letter https://doi.org/10.7554/eLife.56159.sa1
Author response https://doi.org/10.7554/eLife.56159.sa2

## Additional files

### Supplementary files

• Source code 1. Langevin Dynamics Simulation LAMMPS Script. Langevin dynamics simulation script.

• Transparent reporting form

### Data availability

All data generated and analyzed are included in the manuscript and supporting files.

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
