## [Decision Letter]

**Acceptance summary:**

There is growing interest in and a cognitive dissonance between the equilibrium expectation of a single large droplet vs. multiple micro-phase separated droplets that one observes from phase separation that gives rise to biomolecular condensates. Several explanations have been put forth to explain extant observations. Your work provides an intriguing and testable hypothesis based on the dynamics of phase separation of multivalent molecules that conform to a stickers-and-spacers architecture. The concept of two distinct timescales namely, the timescale of diffusion that brings molecules into contact and the reconfiguration times before which the available valence is exhausted provides another explanation for the observation of micro-phase separated droplets as opposed to a single large droplet.

**Decision letter after peer review:**

Thank you for submitting your article "Membraneless organelles are dynamic, metastable long-living droplets formed by sticker-spacer proteins" for consideration by *eLife*. Your article has been reviewed by three peer reviewers, including Rohit V Pappu as the Reviewing Editor and Reviewer #1, and the evaluation has been overseen by José Faraldo-Gómez as the Senior Editor. The following individual involved in review of your submission has agreed to reveal their identity: Alexander Grosberg (Reviewer #3).

The reviewers have discussed the reviews with one another and the Reviewing Editor has drafted this decision to help you prepare a revised submission.

There is growing interest in the topic of "liquid-liquid phase separation" better described as phase separation aided bond percolation as a mechanism by which multivalent macromolecules form biomolecular condensates. The sequence-intrinsic features that give rise to biomolecular condensates can be measured via in vitro reconstitutions and instantiated in numerical simulations. The general framework whereby multivalent macromolecules are mapped onto a stickers-and-spacers architecture is proving to be quite useful for describing the phenomenology and thermodynamic driving forces for phase transitions specifically., phase separation aided bond percolation. Despite these advances, the actual dynamics of phase separation aided bond percolation have been ignored to this point. Are there interesting insights that emerge from a detailed understanding of the dynamics of phase separation aided bond percolation? The answer, according to Ranganathan and Shakhnovich appears to be a resounding yes! Through a combination of multiscale computations, the authors show that multivalent proteins can become dynamically arrested in so-called "micro-phases". As a result, instead of the equilibrium picture of a single, gigantic condensate coexisting with a dispersed dilute phase, one observes the formation of numerous miniature condensates coexisting with the dilute phase. These micro-phases come about because of the rapid manner in which the available valence is exhausted when compared to the slower timescales associated with clusters of molecules diffusing, colliding and coalescing. This gives rise to a dynamical phase diagram that deviates from the equilibrium picture, although the equilibrium free energy landscape still underwrites the driving forces as well as the energetic and / or entropic barriers associated with the dynamics of phase separation aided bond percolation. The results presented by the authors are likely to garner interest given the apparent cognitive dissonance between the equilibrium picture that can be recapitulated in vitro and the observation of multiple smaller facsimiles of condensates coexisting with one other in vivo. This problem has engendered active and animated discussions in the field, and the authors' results as well as their perspectives offer new food for thought.

The manuscript, in its original form, has been reviewed separately by three reviewers. The reviews have been discussed and the general consensus is that this is an interesting and timely contribution that builds on recent successes of the stickers-and-spacers formalism for describing equilibrium phase behavior of multivalent macromolecules. The reviewers noted the importance of the observations that emerge from the simulations. Reviewer 1 is concerned about some bold assertions being made that appear to invalidate the equilibrium picture. It is worth emphasizing that the dynamics are underwritten by the intrinsic free energy landscape and in this work, which does not include sources or sinks, that should be it. That there are interesting dynamics that cause deviations from the equilibrium picture is something that has been reported in a completely different context (please see: https://www.nature.com/articles/s41563-018-0182-6). The consensus is that the authors should articulate the distinction between the equilibrium phase diagram and the dynamical phase diagram, highlight the differences that emerge from the two, and strive to minimize sweeping statements regarding the (in)validity of one over the other, since one (the equilibrium picture) begets the other (the dynamical picture). Reviewer 1 highlighted the crucial need for analyzing density transitions (ignored in the current work) as well as networking transitions (as has been done by Harmon et al. and Choi et al.). This is going to be crucial to understand if the arrested phases are in a distinct regime as far as density is concerned, even though they do appear to be above the percolation threshold as far as connectivity is concerned. The reviewers were unanimous in their dual concerns regarding the number of jargon terms that were introduced throughout the narrative, the interchangeable usage of these jargon terms making things look imprecise, and the assertive use of terms like gel, macroscopic gel, solid-like phases with no quantification of how one can make such assertions absent rheological data (see comments by reviewer 3), characterization of density transitions (see comments by reviewer 1), or experimental data (see comments by reviewer 2). There were concerns about the inaccessibility of the methods descriptions that were raised by reviewer 2 with regard to the description of the Langevin dynamics simulations as well as the Kinetic Monte Carlo simulations. An underlying concern, articulated primarily by reviewer 2, pertains to the artificial imposition of irreversible crosslinks in the LD simulations. The implied concern is that by imposing irreversibility, the observations rather become a tautology. Indeed, while this will not affect the percolation threshold calculated in the mean-field limit, it almost certainly will impact the true estimate of the percolation threshold. A justification for the approach and preferably a comparison to a more realistic scenario in the LD simulations is warranted. Reviewer 2 expressed concerns regarding the hard to follow description of the KMC simulations. A series of concerns were raised and these should be addressed via revisions. Finally, reviewer 3 raised concerns regarding the description of reptation and its usage as a concept in analyzing the simulation results. This is crucial because it sets up the separation of time scales and reviewer 3 is of the opinion that a) the formalism as captured in the mathematics is erroneous and b) an alternate picture might be more relevant to describe the data. Reviewer 1 also highlights the fact that concept of arrested phases is not new. Lee and colleagues have written a series of papers highlighting the possible role of active processes as determinants of micro-phases in cells. The work of Boeynaems et al. highlights the formation of arrested phases due to stable RNA structures. These efforts should be discussed.

As for the title, the assertive declaration is misleading. This work does not actually pertain to a specific biomolecular condensate or membraneless organelle. The title should be modified to better capture the fact that a computational model of multivalent macromolecules modeled using the stickers-and-spacers framework leads to the prediction of long-lived micro-phases. This is the essence of the MS and the title should reflect this point precisely.

Summary:

All in all, while there is strong support for seeing this work published in *eLife*, there are numerous concerns that could be addressed in a revised version with new analysis sans new data, although the inclusion of reversible crosslinks in the LD simulations would be very helpful, even essential. There is concern that the findings will be opaque to the average reader of *eLife*. Therefore, it would be helpful if the authors were to lean on a few colleagues who are primarily molecular or cellular biologists and gain their perspective regarding the overall accessibility of the revised MS and ensure that the message will be clearly understood.

Essential revisions:

The major revisions requested are a) please moderate the rather extreme pronouncements about the irrelevance of the equilibrium picture; b) please analyze density as well as percolation transitions; c) please replace the usage of gelation with bond percolation; d) please streamline the narrative to use a small number of necessary jargon terms and define these up front; e) please redo the LD simulations with reversible crosslinks to obtain a comparative assessment of the pictures that emerge with irreversible vs. reversible crosslinks; f) please reconsider the assertions being made about solid-phases since the results cannot be used to make these assertions; g) please note that Ostwald ripening has been observed in vitro and there are numerous examples, especially from the Brangwynne and Hyman labs showing that a large single droplet will form and coexist with a dispersed dilute phase; h) please include citations to the work of Chiu Fan Lee, Boeynaems et al. and Roberts et al. https://www.nature.com/articles/s41563-018-0182-6; i) please revisit the description and usage of the reptation model, correct the mathematical description to ensure that what emerges has units of time, and ensure that usage of this model is indeed appropriate / accurate; j) please provide a more accessible treatment and an illustrative example of the KMC formalism (details that enable reproducibility are important); k) please provide a summary of the main conclusions, please minimize some of the overstatements and casting aside of previous work, cite the seminal work of Semenov & Rubinstein, and touch base with the work of Huan-Xiang Zhou, especially the recent publication in PNAS.

Reviewer #1:

In this work, the Ranganathan and Shakhnovich present their analysis of dynamical aspects of phase transitions for associative polymers described by a stickers-and-spacers architecture. This framework has been recently adapted to describe the phase behavior of linear and branched multivalent proteins. The central findings in the current work are that multivalent molecules form dynamically arrested micro-phases and that this comes about because of network terminating interactions arising at short time scales that in turn inhibit the growth of the network to give rise to macro-phase separation wherein a single condensate (droplet) defined by a percolated network within the condensate coexists with a dispersed dilute phase. Instead of this equilibrium scenario, the disparate timescales lead to multiple droplets that act as metastable sinks to inhibit the growth of macroscopic phases. In the picture that emerges, the dynamics of phase transitions of stickers-and-spacers systems are governed by a hierarchy of timescales and the separation between distinct timescales gives rise to stable mesophases i.e., metastable phases. This aspect of the authors' finding is a) interesting, b) timely and warrants reporting, and c) reasonably well established in the synthetic polymer literature, although under-appreciated (perhaps even unappreciated) in the biomolecular condensate literature. This work certainly deserves to be published in *eLife*. There are, however, a few key issues that would benefit from careful scrutiny.

1) The authors suggest that the equilibrium state is rarely accessed in vivo or in vitro. This flies in the face of data that have been presented in vitro. Observations of Ostwald ripening and numerous reports of a single large droplet forming in test tubes that coexist with droplet-free dispersed phases have been reported in the literature. There are technical difficulties associated with observing / recording growth dynamics in vitro, especially when the assays are based on fluorescence microscopy, but careful analysis from several labs show that the equilibrium scenario is indeed realized in vitro. One recent example of this is the following paper: https://science.sciencemag.org/content/367/6478/694, but I emphasize that there are many more. What is however true (and not mentioned in the current MS) is that there have been several discussions in the literature about apparent emulsification being observed in vivo. The work of Lee, Jülicher and coworkers introduced the idea of active emulsions – see https://iopscience.iop.org/article/10.1088/1361-6633/ab052b for a review, a direct use of this theory for modeling stress granules http://hdl.handle.net/10044/1/57849, and the original work that introduced the idea of active processes giving rise to essentially uniform distributions of droplets https://journals.aps.org/prl/abstract/10.1103/PhysRevLett.120.078102. Simply stated, there is an active (no pun intended) debate about the mechanisms by which microphases arise. The authors have introduced a dynamical view of this could arise and they suggest that this could be general, which may be the case. However, it is worth emphasizing for accuracy and rigor that the presence / absence of arrested phases will be governed by a combination of factors including what the authors articulate here as well as the quench depth into the two-phase regime (which requires prior knowledge of the equilibrium phase diagram), and the rigidity effects due to prior structure formation – see for example https://www.pnas.org/content/early/2019/03/28/1821038116. The upshot is that it is incorrect to assert that extant data support the contention that the equilibrium scenario is rare nor is it accurate to suggest that the equilibrium phase behavior has zero bearing on "reality". In fact, the equilibrium phase diagram provides the crucial touchstone (the intrinsic free energy landscape) for the dynamical phase diagram. In light of these facts, it would be useful to provide a more nuanced representation of the importance of the equilibrium framework and spell out what the dynamical phase diagram adds to the field (which is a lot). One needs both descriptions, not either / or.

2) In the manuscript, the term phase separation is used often, but the work does not ever analyze an order parameter for density transitions, focusing instead on the clustering alone. This is important because the narrative ends up conflating networking / clustering (gelation i.e., bond percolation), which is what is being quantified, with density transitions i.e., phase separation, which is never actually quantified. The works of Harmon et al., 2017, and Choi et al., 2019, go to great lengths to distinguish between phase separation and bond percolation. In the current context, this would seem to be especially relevant. Are the microphases clusters with densities that are in accord with the dilute phase or do the densities also change? If the latter, then how do the densities (concentrations) within the dilute phases compare to the thermodynamic saturation concentration and how do the densities within the microphases compare with the equilibrium densities within dense phases? These questions can only be answered by deploying two sets of order parameters, one based on networking (which the authors use) and one based on density transitions (ignored in this work).

3) Please note that the phase diagrams presented here are dynamical phase diagrams. They are not equilibrium phase diagrams and to suggest otherwise is misleading. This should be fixed.

4) Finally, on a semantic note: the manuscript is quite long, takes a rather narrow view of the literature, and importantly, introduces too many over-loaded terms (terms used differently in different contexts) and too many distinct terms as well. It would help immensely to streamline the verbiage to define a set of terms and stick with these throughout. Otherwise, I am concerned that the average reader of *eLife* will be quite confused and not dial into the central message of this work.

Reviewer #2:

In this work, the authors use a computational approach to model phase separation of polymers that are comprised of stickers and spacers (similar to that employed in Harmon et al., 2017). By employing computational Langevin dynamics and Monte Carlo simulations, the authors monitor phase separation on biologically-relevant timescales. In so doing, the authors are able to explore the different tunable parameters of phase separation, including non-specific interactions among spacer regions (linkers), specific interactions among sticker regions, linker rigidity, and polymer concentration. By adjusting these parameters, their simulations revealed several interesting points. First, as expected, large clusters (consisting of many polymers) form only at high monomer concentrations. Second, phase separation occurred only when there was high valency with functional interactions among stickers, and not in the absence of such interactions. Third, simulations showed "arrested droplets", i.e. no system-spanning macro-phase separated state was observed at intermediate monomer concentrations. The authors provide evidence that this is due to exhausted free valencies. This observation is tunable based on the interaction strength of the linkers and stickers. The LD simulations also allow for linker rigidity modeling that informed on timescales that impact the timing of kinetically-arrested droplets. Importantly, the authors identify competing timescales for how clusters grow, specifically (1) the initial diffusion encounter, and (2) the reorganization time for the clusters. Next, the authors employed kinetic Monte Carlo simulations to more accurately model interactions that take place. Above, when polymers encountered each other, interactions were irreversible, i.e. once two polymers interacted, they did not come apart again for the rest of the simulation. This is a poor assumption, therefore, the second half of the work employs reversible binding. However, this dramatically changes how the simulation models polymer dynamics, as the polymer is now represented coarsely as a single bead with a variable λ parameter that designates occupied valency.

Overall, the manuscript provides some key insights into the kinetics of droplet assembly, as well as into the material properties of droplets as a function of time. This study further illustrates the usefulness of the stickers and spacers polymer model to obtain the rules for how polymers self-assemble into droplets at relevant timescales. The manuscript needs to be strengthened by clearer language, a better presentation of the results from varying many different parameters that could be tested experimentally, careful summary of the results, and detailed methods so that the general audience can appreciate how the conclusions were reached.

1) There is a need to pictorally represent the “metastable” simulation states when the valencies are filled. This is necessary to illustrate the difference between the macro-phase separated state of a system-spanning single droplet, and what is meant by micro-phase separation of many droplets. Care needs to be taken with the term phase separation propensity as this manuscript quantifies the parameters necessary for phase separation; why not use saturation concentration thresholds (csat), and discuss increasing/decreasing csat? These terms need to be clearly defined for the readership.

2) There is a lot of useful information presented throughout the manuscript of how various parameters tune the kinetics of droplet assembly in the simulation. It is necessary to sum these key points in a conclusion section that is missing from the manuscript (although this reviewer appreciates the testable hypotheses section for useful experimental follow-up experiments). That being said, it would have been very useful to have experimental data to back up some of the simulation results, perhaps via a collaboration with experimental groups on the SH3-PRM system.

3) There are many parts of the text that could be improved with clearer definitions. For example, subsection “Phenomenological kinetic simulations predict microphase separation at biologically relevant timescales” is hard to read, and needs to be broken up so that the results are described succinctly. A second example: I imagine there will be confusion regarding what is considered “gel” in Figure 7 for readers that are familiar with the term in polymer physics, vs. biologists that may confound the definition of “gel” with solid-like material properties.

4) There are interesting results in Figure 5 regarding the relationship between Rg and the concentration threshold for phase separation. These results are particularly interesting for evaluating the contribution of linker regions in multidomain proteins. Are these findings applicable for all phase separating systems? I started thinking about this in the context of proteins that phase separate with increasing temperature (LCST) vs. proteins that phase separate with decreasing temperature (UCST).

5) There is difficulty in reading the Materials and methods section for the kinetic Monte Carlo simulations, and more discussion in the text should be devoted to explaining how the method works, particularly for a general audience. Perhaps an example timestep would be useful here. It is unclear to me how the simulation time is advanced. What is meant by rtotal? Are these the sums of individual rates across all monomers and event types for a given time?

6) Along these lines, please carefully reread the LD methods section. Multiple variables are missing from the text (following Equation 4). This made it impossible for me to understand how LD simulations were executed, and I would like to see this section again.

7) Implications from kinetic MC simulations are that the interplay between interaction strength of specific interactions and valency tune the liquid-like properties of phase separated droplets. Increasing specific interaction strength promotes more solid-like droplets. This is an important point, given recent studies that discuss effects of lysine/arginine residues on droplet material properties and csat. Can the authors comment on this point in their manuscript, even though the sticker-spacer model does not specifically address different amino acid types? This would add value to the findings.

8) Figure 8 needs example simulation results to help the reader understand u1 and u2 differences.

9) Scripts detailing the analysis for the simulations (e.g. largest cluster analysis) are missing and need to be included in the manuscript for clear reproducibility.

Reviewer #3:

This is an interesting paper dealing with a currently very hot subject of liquid-liquid phase separation inside the cell with the participation of appropriate proteins. The heart of the work is a Langevin dynamics simulation accompanied and illustrated by a simple semi-analytical model. The most important novel results deal with the prediction of microphase separation (unlike classical Flory theory predictions) due to the temporal exhaustion of available valences. Importantly, authors claim that this type of phase segregation happens on a biologically relevant time scale.

In my opinion, the main results seem novel, sound, and physically transparent. The model is a reasonable compromise between tractability and approach to reality. I would definitely recommend the work for publication. This general conclusion notwithstanding, I have two issues with the work.

1) One is the use of some loose terms, such as "gel-like" phase; what exactly is it? Even more, there is also "dynamically solid macroscopic gel-like state" as opposed to "liquid-like". What do all these words mean? As far as I understand, authors did not measure shear modulus of their aggregates, so their characterization in terms of solid versus liquid requires clarification. Furthermore, if gel is a percolating covalent network, then it is strictly speaking a solid, at least on the scale above its mesh size. I would welcome a more concise usage of all these words.

2) My second remark is about reptation. Estimate of the time called reptation is very important for the general conclusions of this work. This estimate comes apparently from formula (1) which is very strange. First of all, units do not seem to work in this formula: L = 145 is the number of beads, it is dimensionless, while D_bead_ is the diffusion coefficient; the result cannot be time. While this may seem like a simple typo (e.g., forgotten bead size squared), combined with other issues this one may not be that simple. Second, the real reptation time, unlike formula (1), is not controlled by D_bead_ but by an L times smaller quantity. Third, perhaps the strangest, this strange formula is supported by the reference Feldman, 1989, which is actually a review of the famous textbook by Doi and Edwards. The book would be of course the most welcome source about reptation, but not a review of this book. In fact, my suspicion is that the process in question is not reptation, but some sort of a diffusion along the chain. In any case, authors should really clarify what exactly they mean.

To summarize, I would feel comfortable recommending this work for publication in *eLife* as soon as authors clarify the two issues mentioned above.

---

## [Author Response]

Essential revisions:The major revisions requested are:a) Please moderate the rather extreme pronouncements about the irrelevance of the equilibrium picture.

We appreciate the concerns raised by the reviewers with regards to the lack of acknowledgment of the importance of equilibrium theories. In the revised manuscript, at several places in the Introduction as well as the Discussion, we highlight how the process of phase-separation by multi-valent polymers is driven by the underlying thermodynamic landscape. In the revised manuscript, we emphasize that the current study focuses on potential role of dynamics in hindering the progression of a multi-droplet system to the equilibrium two-state system. We also provide examples from literature supporting the observation of droplet coalescence in support of the equilibrium predictions. The current study sheds light on the importance of dynamics of cluster growth, complementing the existing equilibrium understanding of the phenomenon.

b) Please analyze density as well as percolation transitions.

We thank the reviewer for this suggestion. In the revised manuscript, we introduce a second order parameter to analyze the intracluster density of polymers (normalized by the bulk density). This quantity is analogous to the order parameter “ρ” used to analyze density transitions by Harmon et al^1^. In Figure 1A of the revised manuscript, we plot the intracluster densities and cluster sizes as a function of total monomer concentration (C_mono_). As observed in case of equilibrium lattice simulations by Harmon et al^1^, the density transition shows a non-monotonic behavior as a function of concentration, with large concentrations resulting in system-spanning networks with low densities. In a narrow range of concentrations, we do observe dense clusters. Further, in Figure 5, we show how intracluster density depends on the properties of the linker. In Figure 5B and Figure 5—figure supplement 2, we show how inter-linker interactions can tune the density of polymers within the clusters.

c) Please replace the usage of gelation with bond percolation.

We thank the reviewers for this suggestion. In the revised manuscript, we refer to large clusters with system-spanning networks as a “macrophase” (characterized by low density within the macro cluster), and to a system of coexisting clusters (S_clus_ << N_tot_) as a metastable “micro-phase”. We refrain from using the term gelation in the revised manuscript.

We define these terms in the first subsection of the Results titled “Terminology and Notations used in this study”.

d) Please streamline the narrative to use a small number of necessary jargon terms and define these up front.

We now introduce the conventions and terminology used in the article in the first subsection of the Results titled “Terminology and Notations used in this study”. We also introduce a table listing the physical interpretation and the definition of all variables and order parameters used in the current study.

e) Please redo the LD simulations with reversible crosslinks to obtain a comparative assessment of the pictures that emerge with irreversible vs. reversible crosslinks.

To test whether the early time-scale behavior changes in the presence of reversible interactions, we introduced breakable functional bonds in our model. A comparative assessment of the single largest cluster sizes as well as the distributions for reversible and irreversible crosslinks is now presented in Figure 2B and C. As with the irreversible interaction simulations, even in the presence of breakable interactions, L_clus_ << N_tot_ (Figure 2B) indicating the existence of long-living metastable microphase droplets except for large C_mono_ when we observe a system-spanning network. Critically, we find a coexistence of intermediate cluster sizes with small and large clusters suggesting an increased diversity in cluster sizes at the early stages of droplet assembly even upon introduction of breakable interactions (Figure 2C). However, a more detailed study of impact of bond formation and breakage dynamics on cluster growth is the subject of future work.

The implementation of reversible functional interactions is explained in the “Modeling Specific Interactions” subsection under Materials and methods.

f) Please reconsider the assertions being made about solid-phases since the results cannot be used to make these assertions.

The usage of “solid-like” and “liquid-like” phases in the original manuscript was in reference to the rate at which monomers get exchanged between the condensate and the free medium in our kMC simulations (referred to as exchange times in the manuscript). We do not perform any other detailed assessment of the material properties of these assemblies. We therefore do not use these terms in the revised manuscript. Instead, we call these refer to these phases as slow- or fast-exchange phases in the revised manuscript , based on their mean exchange times in kMC simulations.

g) Please note that Ostwald ripening has been observed in vitro and there are numerous examples, especially from the Brangwynne and Hyman labs showing that a large single droplet will form and coexist with a dispersed dilute phase.

Thanks for pointing this out. In the Introduction section of the revised manuscript, we now include references supporting the in vitro observation of droplet growth via Ostwald ripening and coalescence.

h) Please include citations to the work of Chiu Fan Lee, Boeynaems et al. and Roberts et al. https://www.nature.com/articles/s41563-018-0182-6

In the Introduction and Discussion sections, we now broaden the discussion to include the potential role of active mechanisms as well as the rigidity of the scaffold in promoting/stabilizing a multi-droplet system. We cite the suggested references in the relevant context within the revised text.

i) Please revisit the description and usage of the reptation model, correct the mathematical description to ensure that what emerges has units of time, and ensure that usage of this model is indeed appropriate / accurate.

We agree with the reviewer’s comments about the validity of the reptation based mathematical description of the dimer reorganization timescale. Indeed, a classical reptation model predicts scaling of N^3^ while our formula predicts Rouse-type scaling of N^2^. Extra power of N in reptation model comes from slowing down diffusion in entangled dense polymer solution (melt) which does not play a role in our estimate of time scale of valency saturation in a polymer dimer in solution. However, we feel that this interesting question warrants a separate study. We removed this equation and accompanying discussion from revised manuscript since it is not central to our message here. We plan to return in the near future to careful study of polymer physics of valency exhaustion in dilute solution and report findings in a separate manuscript.

We also cite the sticky-reptation model described by Semenov and Rubinstein in the Discussion section of the revised version. While relevant, it explores somewhat different model of a dense solution while our focus here is on dynamic processes leading to formation of metastable microdroplets from initially dilute solution.

j) Please provide a more accessible treatment and an illustrative example of the KMC formalism (details that enable reproducibility are important).

We now provide additional details of the kMC simulation, and the details of cluster size computations in the Materials and methods section of the revised manuscript. The reactions modeled in the kMC simulations are schematically described in Figure 6. The simulation code for the lattice-kMC simulations will also be deposited in a software repository and will be accessible to the readers.

k) Please provide a summary of the main conclusions, please minimize some of the overstatements and casting aside of previous work, cite the seminal work of Semenov & Rubinstein, and touch base with the work of Huan-Xiang Zhou, especially the recent publication in PNAS.

The summary of the main conclusions is presented in the form of a succinct illustration (Figure 10) as well as the “Testable Predictions” section. We believe that a separate additional conclusions section will lengthen the paper even further and duplicate already existing pictorial conclusion. The work by Semenov and Rubinstein is now discussed in the context of exhausted valencies within clusters in the Discussion section. We also discuss the patchy particle model by Huan-Xiang Zhou’s group in the Introduction section while referring to existing computational studies related to LLPS. We thank the reviewers for bringing these important works to our attention.

Overall, we modified three figures in the main text (Figure 2, Figure 4 and Figure 5) and included a new subsection introducing the key terminology and order parameters used in this study. We include a new subsection in the Materials and methods section, discussing the implementation of breakable interactions. Under Introduction and Discussion in the revised manuscript, we also provide a more elaborate context emphasizing the importance of the existing thermodynamic understanding. Also, to provide a broader view of existing literature, we incorporate references to the possible role of active mechanisms in shaping droplet size distributions and compare predictions from both mechanisms.